# Biosensing with the singular phase of an ultrathin metal-dielectric nanophotonic cavity

Kandammathe Valiyaveedu Sreekanth[1,2], Sivaramapanicker Sreejith[3], Song Han [1,2], Amita Mishra[4], Xiaoxuan Chen[1], Handong Sun [1,2], Chwee Teck Lim[3,5,6] & Ranjan Singh[1,2]

The concept of point of darkness has received much attention for biosensing based on phase-sensitive detection and perfect absorption of light. The maximum phase change is possible at the point of darkness where the reflection is almost zero. To date, this has been experimentally realized using different material systems through the concept of topological darkness. However, complex nanopatterning techniques are required to realize topological darkness. Here, we report an approach to realize perfect absorption and extreme phase singularity using a simple metal-dielectric multilayer thin-film stack. The multilayer stack works on the principle of an asymmetric Fabry–Perot cavity and shows an abrupt phase change at the reflectionless point due to the presence of a highly absorbing ultrathin film of germanium in the stack. In the proof-of-concept phase-sensitive biosensing experiments, we functionalize the film surface with an ultrathin layer of biotin-thiol to capture streptavidin at a low concentration of 1 pM.

[1] Division of Physics and Applied Physics, School of Physical and Mathematical Sciences, Nanyang Technological University, 21 Nanyang Link, Singapore 637371, Singapore. [2] Centre for Disruptive Photonic Technologies, The Photonic Institute, 50 Nanyang Avenue, Singapore 639798, Singapore. [3] Centre for Advanced 2D Materials and Graphene Research Centre, National University of Singapore, 6 Science Drive 2, Singapore 117546, Singapore. [4] Division of Chemistry and Biological Chemistry, School of Physical and Mathematical Sciences, Nanyang Technological University, 21 Nanyang Link, Singapore 637371, Singapore. [5] Department of Biomedical Engineering, National University of Singapore, Singapore 117583, Singapore. [6] Mechanobiology Institute, National University of Singapore, Singapore 117411, Singapore. Correspondence and requests for materials should be addressed to R.S. (email: ranjans@ntu.edu.sg)

It is well-known that enhancement of light absorption in optical structures is only possible when the total combined reflection and transmission are decreased. Over the past few years, perfect absorption of light has become an emerging research area that has been extensively studied in different spectral regions using various optical systems[1–6]. In the case of ideal optical systems, such as for light reflection from a single interface[7], prism-coupled surface plasmon resonance[8], coherent absorption[9], and parity-time metamaterial[10], the absence of reflection (point of darkness) can be achieved at a certain value of the incident angle (the Brewster angle) for specific frequencies and polarization states. However, in practice, the complete suppression of reflection is not possible in such systems due to sample fabrication errors (disorder, inhomogeneity, etc.)

To overcome these technological hurdles and push the limits of complete suppression of reflection, Kravets et al. proposed a concept known as topological darkness[11]. According to this concept, an optical system that follows the Jordan curve theorem[12] can exactly provide complete suppression of reflection at certain incident angles and frequencies. Specifically, this is a phenomenon described in the two-dimensional optical-constant $(n, k)$ plane with a slight restriction on the effective dispersion curve $(n_{\text{eff}}(\lambda), k_{\text{eff}}(\lambda))$ such that the dispersion curve must start and end in two different areas separated by the zero reflection curve[11]. This implies that the point of darkness is topologically protected (disorder of the samples cannot affect the zero reflection point). This concept was first experimentally realized using plasmonic gold nanoarrays fabricated by electron-beam lithography[11]. Since then, different subwavelength nanostructures have been proposed for the realization of topological darkness[13–17]. In a work by Malassis et al., self-assembled plasmonic metamaterials were used to achieve topological darkness[13]. A metallized woodpile structure-based 3D plasmonic crystal metamaterial has been proposed for realizing topological darkness[14]. More recently, Song et al. reported on dispersion topological darkness at multiple wavelengths and polarization states using a three-layered structure where the top layer was nanopatterned[15].

In recent years, optical transduction methods have received considerable attention for the real-time monitoring of biomolecular binding events because they avoid the time-consuming labeling steps[18]. A variety of label-free refractometric sensing devices have been demonstrated using plasmonic nanostructures[19–21], metamaterials[22–26], photonic crystals[27, 28] and whispering gallery mode techniques[29, 30] based on different interrogation schemes (wavelength, angle, intensity, and phase). Among these, phase-sensitive interferometry techniques[31–34] show higher sensitivity compared to sensors based on spectroscopic techniques. Phase-sensitive optical biosensing is one of the most important applications of topological darkness because a singular behavior of the phase in Fourier space is possible at the point of darkness[11, 16]. In particular, an abrupt change in the phase can occur at the topological darkness condition. However, it is important to note that it is not directly possible to use the point of darkness for sensing due to the absence of light and abrupt phase jump at this point. Therefore, the close to zero reflection condition is sufficient for sensing applications. By exploiting this concept, an areal mass sensitivity at a level of fgmm$^{-1}$ with single-molecule sensitivity has been achieved using plasmonic gold nanoarray metamaterial[11]; such sensitivity is much higher than the sensitivity of plasmonic sensors based on spectroscopic techniques. This extreme phase sensitivity was achieved by diffractive coupling of the localized plasmon resonances of gold nanoarrays. In addition, the zero reflection condition attained purely by using the localized surface plasmon resonance also demonstrated much higher phase sensitivity[16]. However, the optical systems reported to date for topological darkness are based on sub-wavelength nanostructures in order to engineer their

effective optical constants to achieve the zero reflection point. Therefore, complex nanopatterning techniques are required for all of these systems, which is not cost effective and restricts the scalability of the samples. Even though perfect absorption has been demonstrated at different spectral regions using various lithography-free metal-dielectric thin-film stacks[35–42], singular phase behavior has not yet been realized.

In this study, we experimentally demonstrate the point of darkness and singular phase using a simple metal-dielectric multilayer thin-film stack at optical frequencies. The proposed system shows a zero reflection point for $p$-polarized light at a specific frequency and incident angle. The point of darkness is observed for one of the modes supported by the system at a specific angle and frequency. The experimental results are fully validated using theoretical simulations. We then propose a fabricated multilayer film for label-free refractometric biosensing. For this purpose, we functionalize the top surface of the stack with an ultrathin layer of biotin-thiol to capture streptavidin. We obtain enhanced phase sensitivity for streptavidin concentrations as low as 1 pM. More importantly, the proposed system does not require nanopatterning to achieve phase singularity, thereby enabling cost-effective large-area multi-channel sensing.

## Results

**Sample fabrication and characterization.** As shown in Fig. 1 (inset of c), the proposed four-layered structure consists of a top silver layer (20 nm), optically thick lossless methyl methacrylate (MMA) as the second layer (522 nm), a highly absorbing ultrathin layer of germanium (Ge) as the third layer (10 nm) and a silver film as the bottom layer (80 nm). These films were deposited on a silicon substrate. Thin films of Ag and Ge were deposited by thermal evaporation, and a layer of MMA was spin-coated on the samples (Methods). The optical constants and thicknesses of the thin films were determined by spectroscopic ellipsometry (Supplementary Fig. 1 and Supplementary Note 1). Field-emission scanning electron microscopy (FE-SEM) images of the Ge surface are shown in Supplementary Fig. 2 and indicate that the thermally evaporated 10 nm thick Ge film has a uniform surface texture and morphology. We performed angular reflectance measurements by using variable-angle high-resolution spectroscopic ellipsometry. To demonstrate the point of darkness, we recorded a pair of ellipsometric parameters such as $\psi$ (ellipsometric reflection) and $\Delta$ (phase shift) as a function of wavelength (400–1000 nm) and angle of incidence (20–90°). To validate our findings, different control samples were fabricated and analyzed. The experimental ellipsometry spectra ($\psi$ and $\Delta$) of the fabricated samples recorded for the incident angle at which minimum $\psi$ ($\psi_{\text{min}}$) was obtained for the longer wavelength mode (second mode) are shown in Fig. 1.

Figure 1a presents the spectra of a two-layered (Ag/MMA) system that supports two cavity modes at 490 nm and 720 nm. These two modes are redshifted and the quality factor of the modes improved significantly by depositing a bottom Ag film with a thickness of 80 nm, as shown in Fig. 1b. For this three-layered system, the $\psi_{\text{min}}$ value could reach as low as 3° for a specific wavelength (835 nm) and angle of incidence (55°), and both modes showed perfect absorption for a wide range of angles of incidence. However, in the case of the three-layered system, singular phase behavior is non-existent for any combination of the thicknesses of the MMA and bottom Ag layers (Supplementary Figs. 3–7 and Supplementary Note 2).

Ellipsometric reflection from the proposed four-layered system is shown in Fig. 1c. In this case, both modes are blueshifted in comparison to Fig. 1b, and the $\psi_{\text{min}}$ value of the longer wavelength mode reached almost zero upon the deposition of a thin Ge layer between MMA and the bottom Ag film. It is also evident that an

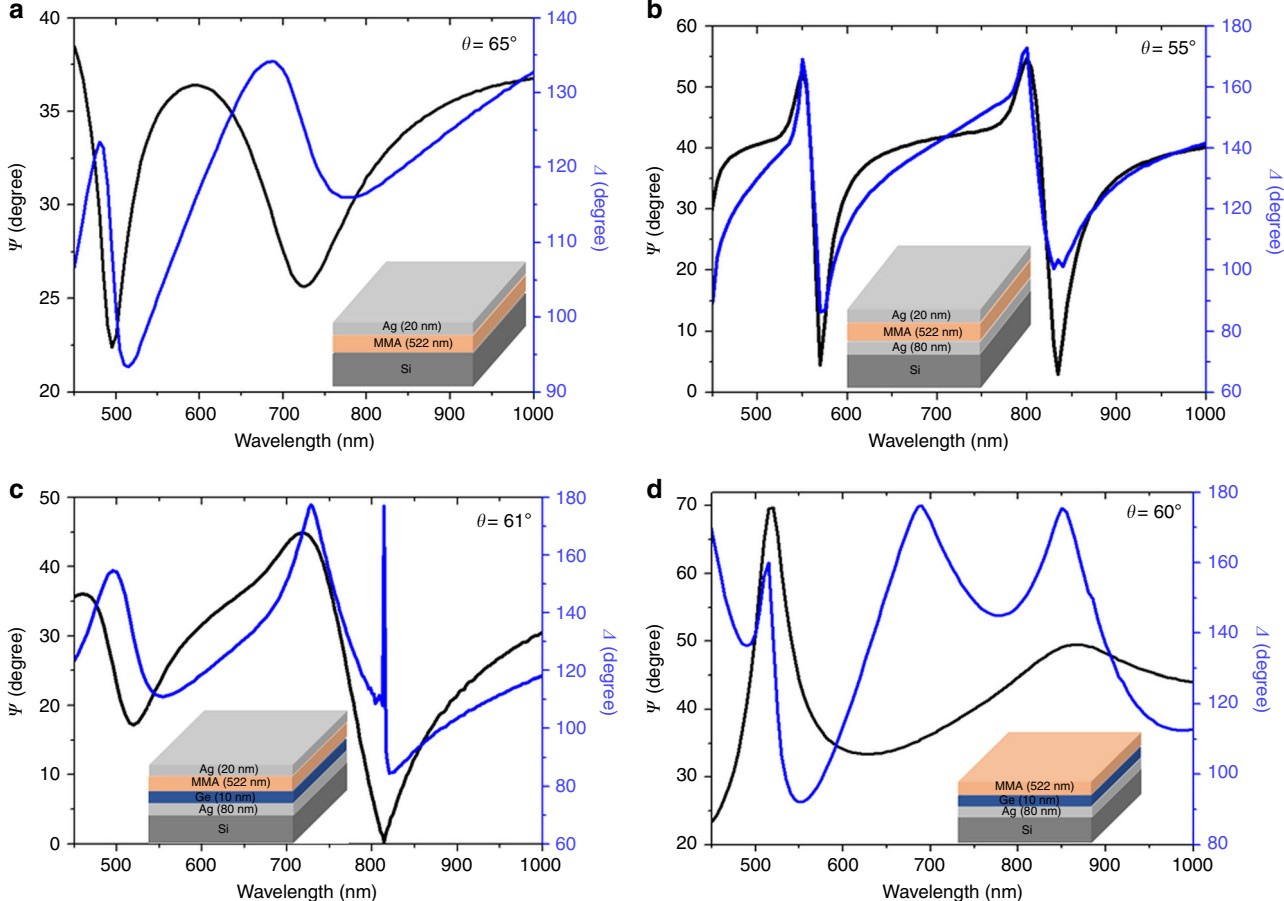

**Fig. 1** Experimentally determined ellipsometry parameters ($\psi$ and $\Delta$) for **a** the two-layered system (Ag/MMA), **b** the three-layered system (Ag/MMA/Ag), **c** the proposed four-layered system (Ag/MMA/Ge/Ag), and **d** in the absence of a top Ag layer (MMA/Ge/Ag). All spectra were plotted for an incident angle at which the minimum $\psi$ was obtained for the longer wavelength mode. All four samples support two cavity modes

abrupt phase change is possible at the reflectionless point where $\psi \approx 0$. This shows that the point of darkness is observable for this system at a wavelength of 814 nm and an angle of incidence of 61° (for more details, see Supplementary Figs. 3–7 and Supplementary Note 2). The principle of achieving a point of darkness and a singular phase in the proposed system is not directly related to the topological darkness phenomenon. In fact, the singular phase is achieved due to the presence of a highly absorbing ultrathin layer of germanium within the stack. The germanium layer with a thickness much smaller than the operating wavelength deposited on a thin silver film with finite optical conductivity rapidly attenuates the incident light. This concept was first proposed by Kats et al.[43] for color filter applications. They demonstrated that a strong interference effect occurs by depositing highly absorbing thin films of germanium on an optically thick gold film. We used a similar concept to realize a point of darkness and a singular phase in the proposed four-layered asymmetric Fabry–Perot cavity.

The spectra appeared to be totally different when the top Ag film was removed from the system (Fig. 1d). The system does not support similar modes, indicating that both observed modes are the cavity modes of the Ag/MMA system and that their behavior can be altered by the deposition of bottom thin films of Ag and Ge with appropriate thicknesses. The $\psi$ spectrum of the system presented in Fig. 1d shows a reverse behavior due to the fact that the s-polarized light provides the minimum reflection at the resonance wavelength compared to the p-polarized light (Supplementary Fig. 8 and Supplementary Note 2).

**Demonstration of a singular phase**. Since only the longer wavelength mode demonstrates the point of darkness, we decided to focus more on this mode in our subsequent investigations. The physical mechanism of achieving a singular phase in the system is solely due to the presence of a highly absorbing dielectric (Ge) thin film (Supplementary Note 2). A thin layer of germanium heavily attenuates the incident light, and the non-trivial phase shift occurring at the interface between Ge and Ag leads to a strong resonant behavior[43]. This is a special case of an asymmetric Fabry–Perot absorber cavity. Asymmetric Fabry–Perot cavities are well-known and have been widely used for different applications[43–45]. Notably, strong absorption resonance occurred when the germanium thickness was much smaller than the wavelength of light[43].

In our case, the thickness $d \approx \lambda/(20n)$ at $\lambda = 814$ nm and $n = 4.2$ for a 10 nm thick Ge film. For the longer wavelength mode, the measured ellipsometry spectra at the angle of incidence of 61° is shown in Fig. 2a as a function of the wavelength. We then selected the point of darkness wavelength as 814 nm and recorded the pair of ellipsometry parameters as a function of the incident angle (Fig. 2c). It can be seen that $\psi_{min}$ and the phase singularity are obtained exactly at 61°. It should be noted that the angular scan provided the lowest $\psi_{min}$ such that a higher phase change at the point of darkness was achieved for the angular scan compared to the wavelength scan. However, the total phase change obtained within three points (above and below $\psi_{min}$) was almost the same. The figure of merit (FOM) of the system based on the ellipsometric parameters can be written as FOM = $\delta\Delta/\delta\psi$, where

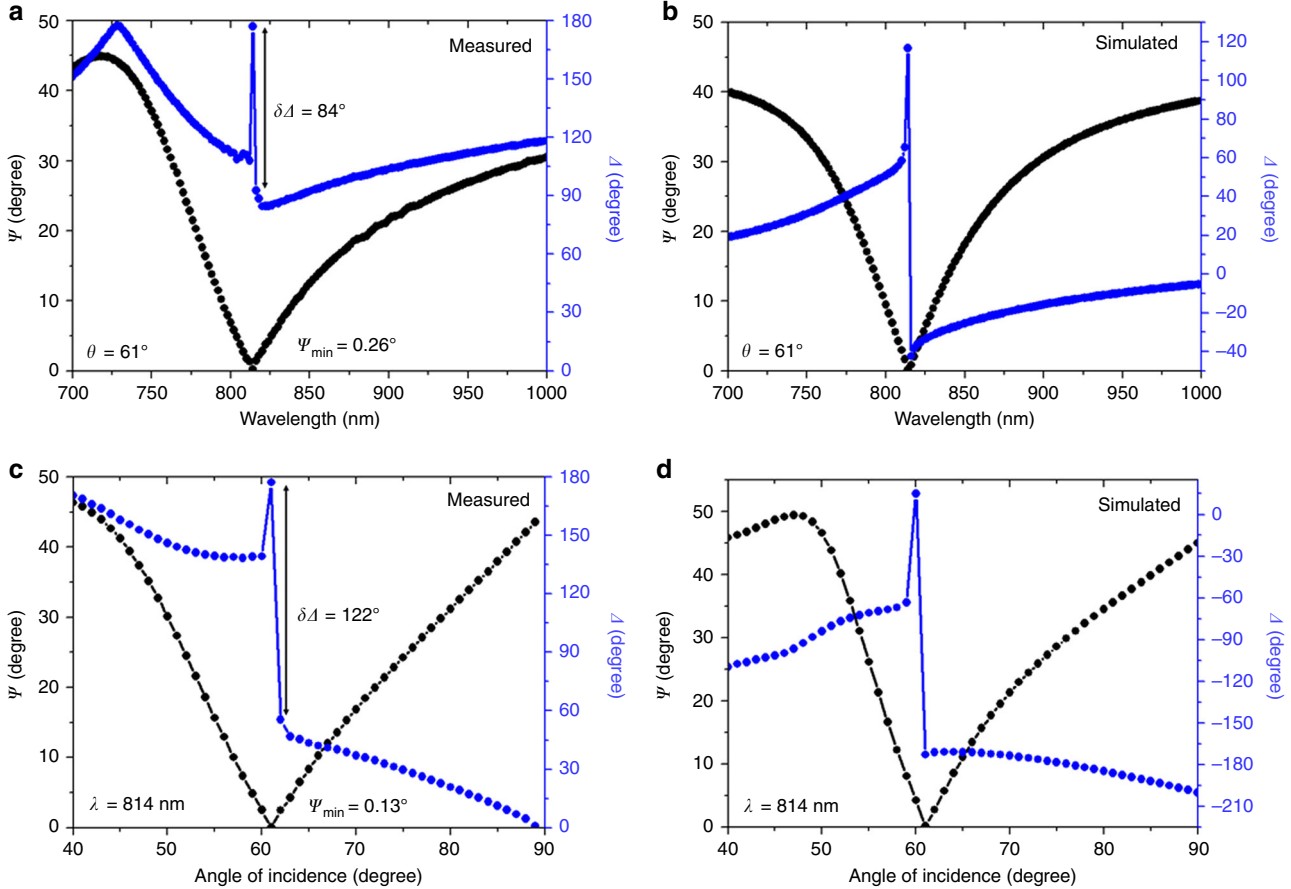

**Fig. 2** Demonstration of the point of darkness and singular phase using the longer wavelength mode. **a** Measured and **b** calculated pair of ellipsometry parameters as a function of wavelength at the angle of incidence of 61°. **c** Measured and **d** calculated pair of ellipsometry parameters as a function of incident angle at a wavelength of 814 nm. The $\psi_{min}$ and phase singularity are exactly obtained at 61° and 814 nm

$\delta\Delta$ and $\delta\psi$ are the change in the phase and amplitude, respectively. This general formula can be simplified as[11] FOM = $1/\psi_{min}$, where $\psi_{min}$ is expressed in radians. The calculated FOM at 61° is 445, where the obtained $\psi_{min}$=0.13° (Fig. 2c). This is the highest FOM reported so far based on the singular phase concept (the previously reported maximum value was 200 in ref. [11]).

The pair of ellipsometry parameters were then calculated by solving Fresnel's equations for an N-layer model using the transfer matrix method (TMM)[34]. In this method, Fresnel's reflection coefficients ($r_{p,s}$) are a serial product of the interface matrix $I_{jk}$ ($j = 0, 1, 2, 3\ldots$ and $k = j + 1$) and the layer matrix $L_j$. That is, $r_{p,s} = \frac{M_{12}}{M_{22}}$, with $M = \begin{bmatrix} M_{11} & M_{12} \\ M_{21} & M_{22} \end{bmatrix}$, $I_{jk} = \begin{bmatrix} 1 & r_{jk} \\ r_{jk} & 1 \end{bmatrix}$ and $L_j = \begin{bmatrix} e^{ik_{zj}d_j} & 0 \\ 0 & e^{-ik_{zj}d_j} \end{bmatrix}$, where $d_j$ is the thickness of the $j$th layer, and $k_{zj} = \sqrt{\varepsilon_j \left(\frac{2\pi}{\lambda}\right)^2 - k_x^2}$, with $k_x$ being the parallel wave vector.

For $p$−polarization, $r_{jk} = \frac{\left(\frac{k_{zj}}{\varepsilon_j} - \frac{k_{zk}}{\varepsilon_k}\right)}{\left(\frac{k_{zj}}{\varepsilon_j} + \frac{k_{zk}}{\varepsilon_k}\right)}$ and for $s$−polarization,

$$r_{jk} = \frac{\left(\frac{k_{zj}}{\sqrt{\varepsilon_j\varepsilon_k}} - \frac{k_{zk}}{\sqrt{\varepsilon_j\varepsilon_k}}\right)}{\left(\frac{k_{zj}}{\sqrt{\varepsilon_j\varepsilon_k}} + \frac{k_{zk}}{\sqrt{\varepsilon_j\varepsilon_k}}\right)}.$$

(1)

Once Fresnel's reflection coefficients for the $p$- and $s$-polarizations are known, the ellipsometry parameter $\psi$ can be obtained

from $\tan\Psi = \left|\frac{r_p}{r_s}\right|$, and $\Delta$ can be calculated from the phase difference between the $s$- and $p$-polarizations $\left(\Delta = \varphi_s - \varphi_p\right)$. The values of $\varphi_s$ and $\varphi_p$ can be extracted from the Fresnel's reflection coefficients of the $s$- and $p$-polarizations, $r_s = |r_s|\exp(i\varphi_s)$ and $r_p = |r_p|\exp(i\varphi_p)$, respectively. The $\psi$ and $\Delta$ values calculated by using this method are presented in Fig. 2b, d as functions of the wavelength and angle, respectively. The calculated results were comparable to the experimental data. In the calculations, the experimentally determined optical constants (Supplementary Fig. 1) of Ag, Ge, MMA, and Si were used, and the thicknesses of the thin films were varied slightly from the measured values to obtain a comparable fit. The layer thicknesses used in Fig. 2b were 21 nm, 538 nm, 11 nm, and 80 nm for the top Ag, MMA, Ge, and bottom Ag, respectively, and the corresponding thicknesses used in Fig. 2d were 18 nm, 532 nm, 9 nm, and 80 nm. We further analyzed the point of darkness conditions by changing the Ge layer and Ag layer thicknesses (Supplementary Figs. 9, 10). We found that the phase change at the reflectionless point is extremely sensitive to the Ge layer thickness.

It is a well-known phenomenon in Brewster angle microscopy that the reflection becomes extinct when $p$-polarized light is incident on a semi-infinite transparent medium. However, the Brewster angle condition is absent if the medium is an absorbing medium with a complex refractive index. Since our designed optical system consists of absorbing media such as silver and germanium, in principle, the Brewster angle microscopy condition does not exist anymore. To verify that the point of darkness

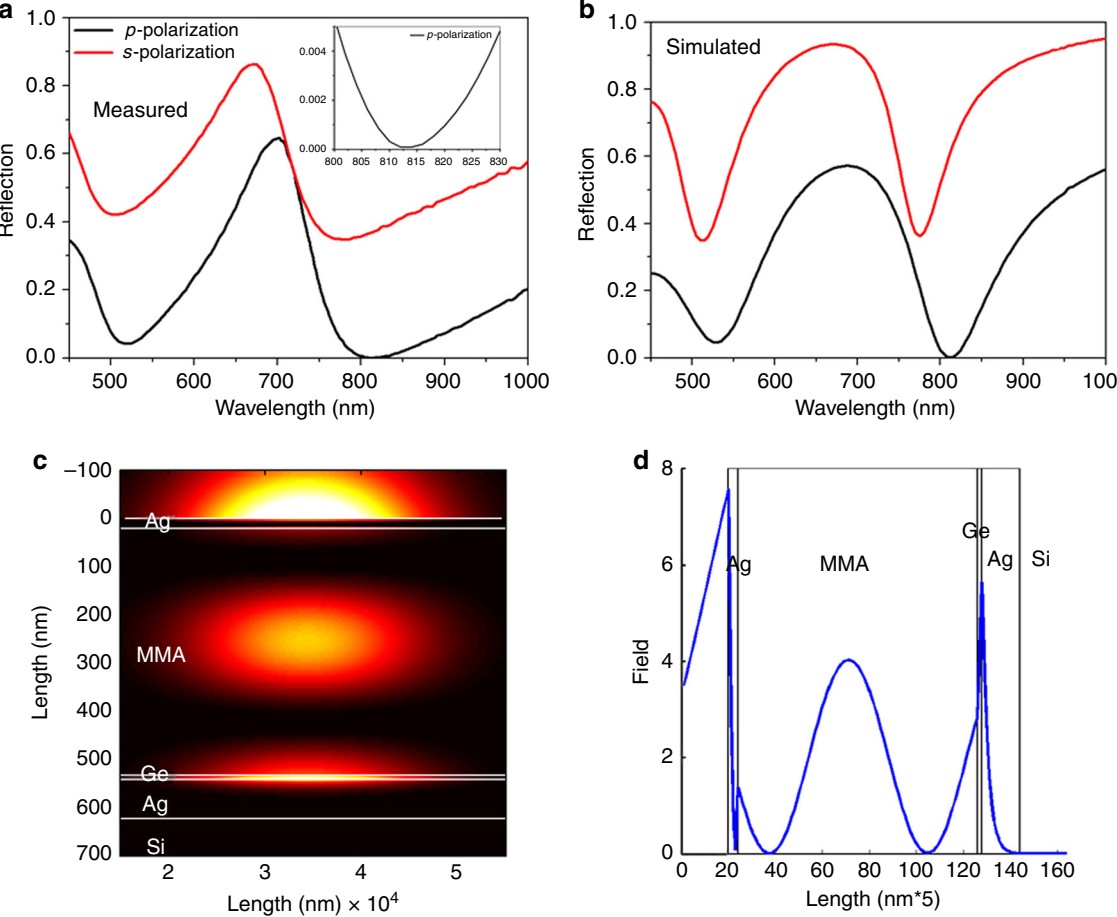

**Fig. 3** Reflectivity spectra and field distribution. **a** Experimental and **b** calculated reflection spectra for *p*- and *s*-polarization at a 61° angle of incidence. The obtained reflected intensity at 814 nm is 0.001%. **c** Calculated field distribution along the four-layered system at the point of darkness condition (814 nm and 61°), and **d** its magnitude as a function of depth along the multilayer

is obtained at the Brewster angle (pseudo), we recorded the reflectance spectrum for the *p*- and *s*-polarizations separately. The measured and calculated ($R_{p,s} = |r_{p,s}|^2$) reflectivity spectra at the angle of incidence of 61° for both polarizations are shown in Fig. 3a, b, respectively. Both reflectivity spectra show that *p*-polarized light provides exactly zero reflectance at 814 nm, confirming that the incident angle (61°) at which zero reflectance was obtained is the Brewster angle. In particular, the *p*-polarized reflection decreased and the *s*-polarized reflection increased after the introduction of an ultrathin Ge layer into the sample (for more details, see Supplementary Note 2).

We further performed field distribution simulations to show the important role of Ge in the proposed system. The electromagnetic field distribution along the system was simulated using TMM. A two-dimensional field map obtained at the point of darkness (wavelength of 814 nm and angle of incidence of 61°) and the corresponding line profile are shown in Fig. 3c, d, respectively. It can be seen that the field intensity remained tightly confined in the thin Ge layer at the point of darkness. This result serves as direct evidence of the physical mechanism for observing the point of darkness where the Ge layer heavily attenuates the incident light. Supplementary Figure 11 illustrates the field distribution obtained for the first mode (520 nm) at the angle of incidence of 61°. Evidently, the confined field in the cavity (MMA layer) was almost the same as that in the Ge layer. Since no significant field confinement is obtained in the Ge layer, the first mode did not show a point of darkness at the wavelength of 520 nm and the minimum reflection angle (61°).

**Phase-sensitive biosensing**. Since the top layer of the proposed four-layered system is a silver thin film, a strong decaying field along the superstrate is possible, thereby enabling the development of a phase-sensitive label-free refractometric biosensor. As mentioned above, wavelengths close to the zero reflection point were sufficient for biosensing so that a slight increase of the reflection due to the immobilization of the ligands on the top metal surface does not degrade the phase sensitivity of the sensor. Therefore, we developed a proof-of-concept biosensor platform to detect streptavidin (52.8 kDa) by immobilizing the top silver surface with a very thin layer of ultra-small molecular weight (244 Da) protein such as biotin.

The immobilization process is as follows: first, we prepared biotin disulfide using a modified synthetic protocol followed by the preparation of biotin-thiol (Methods). Surface functionalization was further confirmed using X-ray photoelectron spectroscopy analysis by tracing elemental sulfur and nitrogen residues post modification (Supplementary Note 3 and Supplementary Figs. 12 and 13). In the next step, the sensor was immersed in different concentrations (0.1 mM, 1 mM, and 10 mM) of the biotin-thiol solution overnight. Then, the sensor was washed with phosphate-buffer saline (PBS) to remove the weakly attached and unbound biotin-thiol and was dried in flowing nitrogen. The film was then used in this state for ellipsometry characterizations.

The measured spectra ($\psi$ and $\Delta$) of the 1 mM biotin-thiol covered sample are shown as the black curves in Fig. 4a, b. It can be seen that the minimum reflection (a point of non-complete darkness) was obtained at a higher incident angle (65°) and that

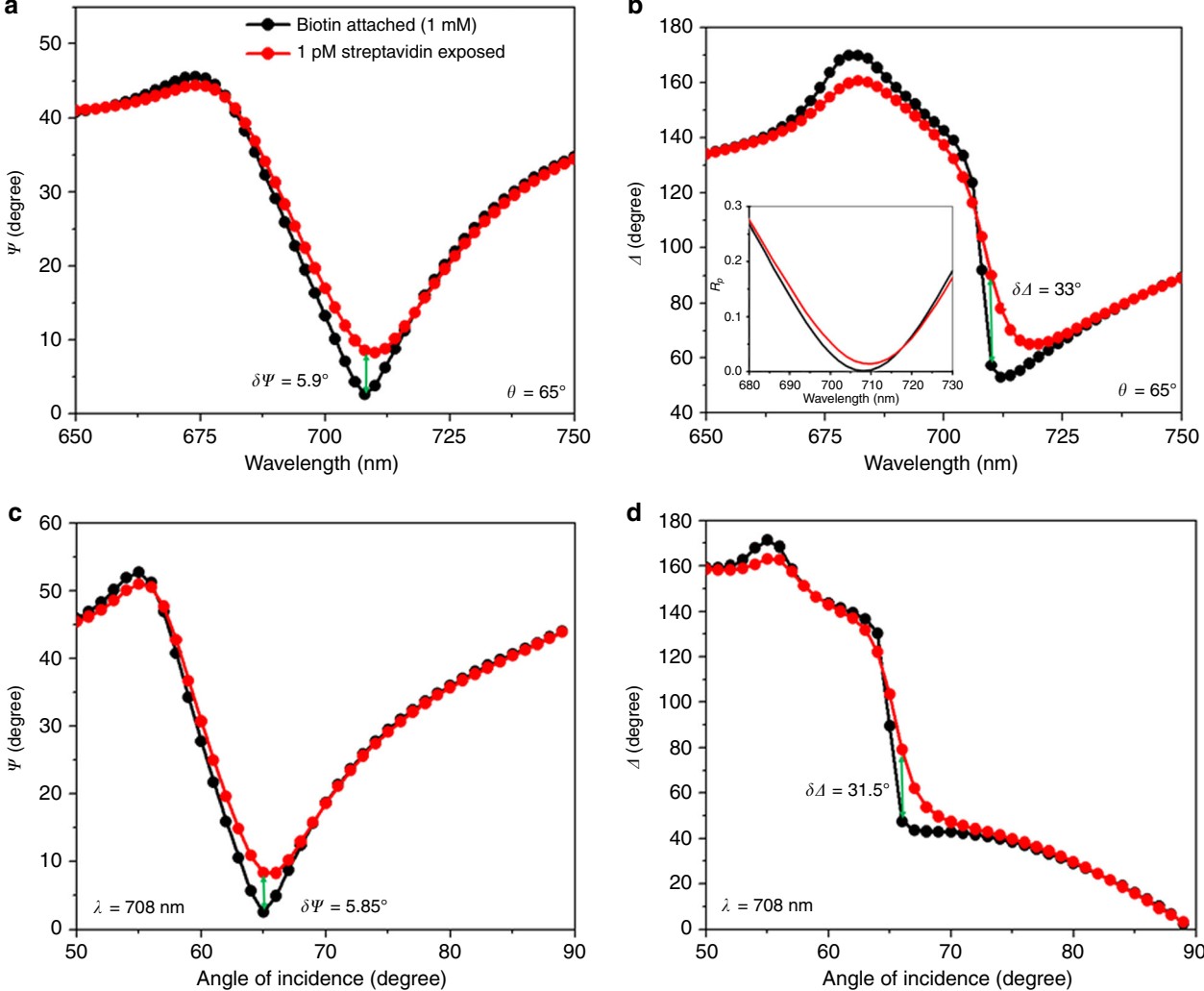

**Fig. 4** Experimental biosensing. Ellipsometry parameters **a** $\psi$ and **b** $\Delta$ versus wavelength before (black curve) and after (red curve) exposure to 1 pM streptavidin at 65°. Ellipsometry parameters **c** $\psi$ and **d** $\Delta$ versus incident angle before (black curve) and after (red curve) exposure to 1 pM streptavidin at 708 nm. The sample surface was functionalized using 1 mM biotin-thiol in PBS. The recorded maximum phase change for the 1 pM streptavidin concentration was 33° at a 65° angle of incidence

the wavelength corresponding to $\psi_{\min}$ was drastically blueshifted to 708 nm. This phenomenon occurred because the surface roughness of the sample was slightly increased due to the presence of an ultrathin biotin-thiol layer. The thickness of this layer increased with an increase in the biotin concentration (Supplementary Notes 4 and 5). We then fixed the minimum reflection wavelength at 708 nm and recorded the pair of ellipsometry parameters $\psi$ and $\Delta$ as a function of the incident angle, which are shown as the black curves in Fig. 4c, d, respectively. To detect heavier molecules such as streptavidin, biotin-thiol covered samples were exposed to 1 pM streptavidin in PBS for 2 h, followed by rinsing with PBS to remove the non-specifically bound streptavidin molecules. This was performed without removing the sample from the ellipsometer stage (in order to keep the same incident beam position on the sample so that we could extract the correct shift).

The ellipsometry parameters recorded after exposure to 1 pM streptavidin are shown as the red curves in Fig. 4a, b for the wavelength scans and in Fig. 4c, d for the angular scans. The p-polarized reflection spectra are also shown in the inset of Fig. 4b. It is evident that a significant phase shift was obtained at $\psi_{\min}$ (wavelength and angle) compared to the corresponding $\psi$ and

wavelength change. We recorded the maximum $\Delta$ change ($\delta\Delta$) of ~33° at the angle of incidence of 65° for the 1 pM streptavidin concentration. Hence, it was confirmed that the obtained phase sensitivity was mainly due to the specific adsorption of streptavidin on the sensor surface (refractive index changes due to specific binding of streptavidin) by conducting an experiment in which the results for different concentrations of biotin attached samples were compared (Supplementary Figs. 14–17 and Supplementary Notes 4 and 5). It is important to note that the phase sensitivity strongly depended on the $\psi_{\min}$ value.

Since the illuminated beam diameter on the sample was ~1 mm², the estimated streptavidin molecule ($N_{\max}$) concentration at the 1 pM concentration was ~$11.8 \times 10^5$ M$^{-1}$. Therefore, the calculated device sensitivity ($\delta\Delta/N_{\max}$) is $2.8 \times 10^{-5}$ degrees, which is one order of magnitude higher than the spectral sensitivity, $\delta\lambda/N_{\max}$ (Supplementary Note 6). Since the higher phase sensitivity of the plasmonic gold nanoarray was due to the super narrow mode plasmonic resonance[11], it may be possible to improve the sensitivity of the proposed sensor further by narrowing the linewidth of the cavity mode (by achieving a high-quality factor mode). In fact, we have achieved a narrow linewidth mode by replacing the Ge layer with a phase change

material such as $Ge_2Sb_2Te_5$ (Supplementary Fig. 18 and Supplementary Note 7).

## Discussion

The abrupt phase change at the point of darkness has previously been demonstrated based on the concept of topological darkness, which requires complex lithography techniques for the fabrication of nanostructures to engineer the effective index. In this work, we have experimentally realized a route to achieve an abrupt phase change at the point of darkness in a lithography-free four-layered metal-dielectric system at the Brewster angle. The proposed optical system showed a large phase change at the point of darkness due to the presence of a highly absorbing thin layer of germanium. Since the top layer is a silver thin film, the oxidation of the silver film could slightly increase the $\psi_{min}$ (Supplementary Note 8 and Supplementary Figs. 19 and 20). The oxidation issue can be resolved to a large extent by using capping layers such as platinum or gold (3–5 nm thickness) or optimized-thickness bimetallic silver/gold layers (gold as an outer layer)[46].

The observed phenomenon could have applications in devices including biosensors and perfect absorbers. We further demonstrate the potential of this thin-film stack for label-free refractometric biosensing due to the abrupt phase change at the reflectionless point. The sensor platform showed enhanced phase sensitivity by detecting relatively heavier molecules such as streptavidin at a 1 pM concentration. As can be clearly seen from Fig. 3c, d, the field decay length into the sensing medium at the point of darkness condition was a few hundreds of nanometers, which is comparable to that of propagating surface plasmon polaritons (SPP)[25]. Therefore, using the singular phase concept in metal-dielectric film stacks, it should be possible to obtain a higher refractive index sensitivity that is comparable to that of SPP sensors. At the same time, the narrow resonance (high-quality factor) mode is equally important for achieving a higher refractive index sensitivity for this system. Since the darkness point depends strongly on the phase sensitivity of the device, this sensor is more suitable for the detection of small molecular weight biomolecules. More importantly, the proposed system is a lithography-free nanophotonic platform, which can be useful for realizing large area biosensors with multi-channel sensing capabilities. The fabrication costs are still a major concern for the full realization of advanced point-of-care tools. Therefore, the proposed nanophotonic thin-film platform could provide a practical solution for this problem.

## Methods

**Sample fabrication**. The multilayer stacks were produced by the sequential deposition of Ag, Ge, MMA, and Ag layers on a clean silicon substrate. Ag and Ge films were deposited by thermal evaporation (Oerlikon Leybold vacuum) of Ag and Ge pellets at deposition rates of 0.2 As$^{-1}$ and 0.1 As$^{-1}$, respectively, and a common base pressure of $<5 \times 10^{-6}$ mbar. However, the MMA copolymer (8.5MMAEL 11, MICROCHEM) was spin-coated at different r.p.m. such as 2000, 3000, and 4000 and baked at 160 °C for 5 min.

**Ellipsometry characterizations**. Variable-angle high-resolution spectroscopic ellipsometry (J.A. Woollam Co., Inc., V-VASE) was used to determine the thicknesses and optical constants of thin films. The ellipsometry parameters ($\psi$ and $\Delta$) and the polarized reflectivity spectra as a function of the excitation wavelength were acquired using the same instrument with a wavelength spectroscopic resolution of 1 nm. In the angular scan measurements, the angular resolution of the ellipsometer was set to 1°.

**Surface chemistry**. Preparation of biotin disulfide (Product 1): In a 100-mL round bottom flask equipped with a magnetic stir bar, biotin (0.28 g, 1.15 mmol) and L-cystine dimethyl ester (0.20 g, 0.06 mmol) were dissolved in dry dichloromethane (DCM) (15 mL) under constant stirring at 600 r.p.m. N-methylmorpholine (0.13 mL, 1.18 mmol) was then added to the above mixture, and the mixture was cooled to −10 °C in a salted ice bath (ice + sodium chloride to attain a −10 °C temperature). N-(3-dimethylaminopropyl)-N′-ethylcarbodiimide hydrochloride (EDCl)

(0.18 g, 1.16 mmol) was then added to the reaction mixture over 5 min and stirred continuously for another 12 h. The reaction progress was monitored by thin-layer chromatography, and upon completion, the reaction mixture was quenched using cold 1 N hydrochloric acid (HCl) (15 mL) and extracted with DCM. The extract was washed twice with DCM (2 × 20 mL) to ensure the complete extraction of the organic phase. Then, the combined organic phase was washed with brine and dried with anhydrous sodium sulfate ($Na_2SO_4$), and the solvent was removed under reduced pressure to obtain white solid Product 1 (0.41 g, 69%). MS (ES): $m/z = 721.5$ [M + 1]$^+$. Preparation of biotin-thiol (Product 2): To a stirred solution of Product 1 (0.15 g, 0.21 mmol) in anhydrous dimethylformamide (4 mL) at ambient temperature under argon atmosphere, tris(2-carboxyethyl) phosphine hydrochloride (0.06 g, 0.25 mmol) was added and dissolved in water (0.4 mL). After 20 h, the reaction mixture was quenched with saturated sodium bicarbonate solution ($NaHCO_3/H_2O$) and extracted with ethylacetate (EtOAc) repeatedly three times. The combined organic layer was washed with brine and dried with anhydrous sodium sulfate ($Na_2SO_4$), and the solvent was removed under reduced pressure to obtain Product 2 (0.07 g, 93%) as a colorless oil. The $^1H$ NMR (nuclear magnetic resonance spectroscopy) and mass spectrometry data of Product 2 are shown in Supplementary Note 3.

**Immobilization of biotin-thiol on the sensor surface**. Samples were soaked in solutions containing different concentrations (0.1 mM, 1 mM, and 10 mM) of biotin-thiol and kept undisturbed for 12 h. The surface was washed using PBS buffer to remove the weakly attached and unbound biotin-thiol, dried in flowing nitrogen and used as such for further analysis. Streptavidin solution (1 pM) was prepared in 10 mM PBS.

**Numerical simulations**. The pair of ellipsometry parameters, reflection spectra, and field map were numerically computed with MATLAB.

**Data availability**. All data generated or analyzed during this study are available from the corresponding author upon reasonable request.

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

## Acknowledgements

The authors (K.V.S., S.H., and R.S.) acknowledge the Singapore Ministry of Education (MOE) Grant Nos. MOE2011-T3-1-005 and MOE2015-T2-2-103 for funding this research. S.S. and C.T.L. also acknowledge support from the National Research Foundation, Prime Minister's Office, Singapore under its medium-sized centre programme, Centre for Advanced 2D Materials. We thank Mr. Deblin Jana (NTU-Singapore) and Dr. P. Borah (University of Tokyo, Japan) for their support with XPS analysis.

## Author contributions

K.V.S. conceived the idea and designed the research. K.V.S. fabricated and characterized the samples, performed sensing experiments and carried out calculations and simulations. S.S. designed the organic synthesis protocols and conducted necessary surface modifications and surface characterization experiments. S.S. and A.M. performed the synthesis and characterization of the ligands for the surface functionalization. S.H. helped with simulations. X.C. helped with the characterization of the samples. K.V.S., S.S., and R.S. wrote the manuscript based on the input from all authors. R.S., C.T.L., and H.S. supervised the project. All authors analyzed the data and discussed the results.

## Additional information

**Competing interests:** The authors declare no competing financial interests.

