## [Peer Review File · Nature Communications]

Reviewers' comments:

Reviewer #1 (Remarks to the Author):

COMMENTS:

The manuscript reports on topological darkness and phase singularity of a simple metal-dielectric multilayer thin film stack. Using the principle of asymmetric Fabry-Perot cavity, the authors demonstrate abrupt phase change at the (nearly) zero reflection point for a specific angle of incidence and wavelength. By exploiting sensor's concept, it was achieved mass sensitivity for relatively heavier molecules such as streptavidin at a level of 1pM concentration due to abrupt change of phase at the reflectionless point. The authors show a nice agreement between the experiment and the theory. The material is well structured and clearly written. The measurements are well made. The paper is of interest to the photonics community, especially to the researchers interested in biological sensing.

The manuscript could be considered published after some revisions:

1) The authors claim in the introduction and final part of manuscript that "this could make its application to ultrathin layer of biotin-thiol to capture streptavidin". It is not immediately clear how and why this would work, taking into consideration how biological molecules would bind to the sample, especially to thin top Ag layer (18 nm) which is not the best for bio-functionalization and suffers the chemical instability due to the oxidation with time.

2) In the manuscript the authors explain the relevance of their work in the context of phase sensitive a simple metal-dielectric multilayer thin film stack. The so-called "topological darkness," is a known property of classical electromagnetism. It is the generalization of the Brewster's angle in the complex plane and multi-layered systems and has been observed many times in the past. Could authors suggest the way (recipe) to achieve the super narrow resonances for such simple layered structures?

3) Usually thin metallic and semiconductor films are not continuous below $\sim 10-15$ nm under thermal evaporation conditions. There will be islands on the surface of film, which can be seen in high resolution SEM or TEM. Could authors provide structural characteristics for Ge film; this will impact the validity of work.

4) It is mentioned that data is acquired from 400nm-1000nm. Nevertheless, the data shown is only between 700 and 1000nm (Fig.3) for spectral dependence of reflections, while the data (Fig. 1) for ellipsometric functions is shown from 400 to 1000nm. I suggest being consistent and using the same wavelength scale in both figures for clarity.

5) Authors should provide the detailed and deep explanation for Figs. 1d and . 3a: from 800-950 nm shown in Fig. 3(a) and (b), why the absolute value in R_p is zero, but for Ψ approaches are non-zero? Although the data is very compelling, Ψ indeed approaches close to zero, the data does not support Ψ reaching absolute zero in the experiment.

6) Measured ellipsometric parameters Ψ and Δ for biotin-thiol covered sample are impressionable because of the ellipsometer functions are rather physical. Meaningful would be the p-polarized reflection spectra for those findings, rather than Ψ .

7) In Fig. S1 authors display the measured optical constants (n and k), using Woollams' ellipsometry, of thin films: Ag, Ge, MMA and Si. These data were taken for thick film Ag(50nm), Ge (100nm) but in

the four-layered system in the case of darkness condition the thickness of corresponding film is smaller: Ag (30nm), Ge(10nm). It is well known that the complex refractive index of thermally deposited films strongly depends on thickness of film (for small thickness <50nm) and conditions of growing. It is not clear why authors did not measure the films with real thicknesses of metal-dielectric multilayer thin film stack.

Another question about data at Fig. S1, why for Ge film authors present the real and imaginary parts of the refractive index but for Si film only the real part, n?

Reviewer #2 (Remarks to the Author):

This is a well written manuscript which reports advances in ultra-sensitive biodetection on a multilayer structure without nanofabrication.

Comments

1. Novelty: As the authors point out, multilayered metal-dielectric perfect absorbers similar to the ones reported here and that are not directly related to topological darkness and do not require nanopatterning have been studied in the literature before, in theory and in experiment, recent examples are DOI:10.1364/OE.22.008339 | OPTICS EXPRESS 8339 (theory + exp); S. Shu, Z. Li, and Y. Y. Li, "Triple-layer Fabry-Perot absorber with near-perfect absorption in visible and nearinfrared regime," Opt. Express 21(21), 25307–25315 (2013). The impact of their work therefore rests on the application of these structures in biosensing. Here, I would question if pM detection of Streptavidin is a breakthrough that warrants publication in a Nature Journal.

2. It is confusing that the authors first introduce the importance of topologically protected darkness for sensing applications but then state that in fact their multilayer system is not related to this phenomenon: "principle of achieving point of darkness in the proposed system is not directly related to topological darkness phenomenon". Can the authors please clarify whether their results have any relevance for the "topological darkness phenomenon" and to which degree their point of darkness is topologically protected?

3. Data: All sensing experiments should be repeated at least three times. Best would be to plot all three results or alternatively, error bars should be included in the data plots.

4. The authors introduce the state of the art in biosensing in terms of device sensitivity reported in fg/mm² area sensitivity. In their own experiments, they report the detection of streptavidin from a solution with pM concentration. From this concentration measurement it is not possible to infer their device sensitivity and compare with the state of the art. Related to this point, picomolar detection sensitivity for streptavidin depends on a number of parameters, including the number density of biotin receptors at the sensor surface, percentage unspecific binding, incubation time, refractive index changes due to exchange of buffer solutions, etc; parameters most of which are not reported in this manuscript.

5. The authors state: "which is much larger than the sensitivity of plasmonic sensors that works based on spectroscopic techniques: " which spectroscopic techniques out of the many possible are the authors specifically referring to? To my knowledge plasmon resonance-based sensors have detection sensitivity that are now reported at the level of single atomic ions, Nature Photonics 10, 733–739

6. Endpoint measurements are always difficult to judge: it remains unclear which part of their detection signal for streptavidin results from bulk refractive index change, unspecific adsorption,

temperature drifts etc. Perhaps more convincing would be a plot for the real time detection of streptavidin, plotting the signal change as the protein accumulates at the sensor surface over time, and including a baseline measurement before the protein is added to the sample solution?

7. It would also be nice have a quantitative interpretation of their sensing signal: Can the authors not determine the effective thickness of the adsorbed streptavidin layer? Related to this: why is there a blue shift of the spectrum due to the biotin layer, if understood correctly?

8. Functionalisation of the surface with biotin seems to have effect on zero point darkness measurements, degrading performance of the sensing system. Can the authors comment?

9. It seems that the figure of merit they apply for zero reflection measurement does not directly apply for sensing experiments since "Therefore, close to zero reflection condition is sufficient for sensing applications". Can the authors clarify?

Reviewer #3 (Remarks to the Author):

I have carefully read the manuscript from Kandammathe Valiyaveedu Sreekanth et al. entitled "Realization of singular phase and point of darkness in optically thin metal-dielectric films for ultrasensitive biosensing applications". This paper uses the concept of "point of darkness" for sensing purpose, and realize the "point of darkness" using lithography-free thin film structures. The idea of this work is interesting, but I feel the overall quality of this work does not meet the standards of Nature Communications.

The major issues I have with this manuscript are as follows:

#1 The authors missed quite a lot of relevant existing work in the literature. In the abstract, it is said (Page 2 line 26-28)

"Here, we propose and experimentally demonstrate an entirely different approach to realizing point of darkness and phase singularity using a simple metal-dielectric multilayer thin film stack."

As well as in the Discussion part (Page 11 line 235-237):

"The point of darkness so far has only been demonstrated based on the concept of topological darkness for which requires complex lithography techniques for fabricating nanostructures to engineer the effective index."

These arguments are either too strong or not right. In fact, there are a lot of work been done in the literate to realized perfect absorption just using thin film structures. For example:

[1] Kats, Mikhail A., et al. "Ultra-thin perfect absorber employing a tunable phase change material." *Applied Physics Letters* 101.22 (2012): 221101.

[2] Li, Zhongyang, Serkan Butun, and Koray Aydin. "Large-area, lithography-free super absorbers and color filters at visible frequencies using ultrathin metallic films." *ACS Photonics* 2.2 (2015): 183-188.

The statement from this work "has only been demonstrated..." is not correct at all. The authors need to change their statement in this work, also mention the relevant work on perfect absorption using thin film structures.

#2. There is a logic flaw in the analysis of the reason for the perfection absorption. In Figure 1, the authors attribute the minimum of reflection to the high absorption of the germanium layer. I agree with the authors that in their specific design, germanium indeed contributes significantly to the perfect absorption of their thin film structures. But this statement is not general at all. The insertion of the germanium layer is not necessary in order to realize perfect absorption. In fact, one can achieve

perfect absorption using the similar structure shown in Fig. 1(b), but using appropriate combination of silver and dielectric thickness (See Ref [2] above). The authors should emphasize this point in their paper.

#3. No one expects perfect agreement between theory and experiment, due to fabrication issues. But the agreement between theory and experimental result of this work is not "very well" (page 9 line 117) at all. For example, in Fig. 2 (a) and (b), the difference in the phase difference Δ is 40deg (experiment) VS 150deg (theory). Also the change of phase difference profile before 875nm is quite difference. Same thing in Fig. 3 (a) and (b), the agreement between theory and experiment is not good enough. In fact, considering the fact that this work is just thin film stacks, no complicated lithography is involved, one expects a much better agreement between simulation and measurement than what is shown here.

Following are some minor issues that the authors might consider when improving their manuscript:

#1: Page 5, line 95, it is stated that

"Brewster angle condition does not exist if the medium is an absorbing medium with complex refractive index."

While it is also stated that later (Page 9, line 181):

"To verify that the point of darkness is obtained at the Brewster angle, ..."

These two statements are not consistent. This needs to be addressed.

#2: Page 6, line 126-127:

"The experimental ellipsometry spectra (ψ and Δ) of all the fabricated samples at the minimum reflection angle are shown in Fig.1.

"

The ellipsometry spectra are wavelength dependent. What is minimum mean? Across all wavelength? This needs to be clarified.

#3: A quantitative discussion on the requirement on magnitude on ψ should be provided. How close to zero does ψ needs to be in order to see an abrupt phase change? As the authors stated in the manuscript, sensing cannot be done at exact location where reflection goes to zero. Rather most devices work on a close point where reflection is not exactly zero. But how small exactly it needs to be in order to observe this? I think a discussion on this would be helpful.

#4: page 9, line 175:

"the thickness of thin films were slightly varied from the measured values, in order to get a good fit." These numbers regarding the thickness of each layer used in simulation should be provided.

#5: page 9, line 178-179

"We further analyzed the point of darkness condition by changing the Ge layer and Ag layer thicknesses".

The dependence on the performance on the Ge layers is shown in the supporting materials, but the dependence on the Ag layer thickness is not shown. How the darkness point changes as the Ag layer thickness is varied?

#6. Thin film based sensor has the advantage of lithography free. But one big advantage of sensor based on nano structures is that the decay length of the localized plasmon mode is much smaller than propagating plasmon mode (20-30 nm vs few hundred nm). Looking from Fig 3 (d), the field decay length here is on the order of few hundred nm. I think some discussion on this regarding to sensing would be helpful.

List of changes made in the revised manuscript based on the reviewer's comments

Reply to Reviewer-1

Reviewer#1

Comment: *The manuscript reports on topological darkness and phase singularity of a simple metal-dielectric multilayer thin film stack. Using the principle of asymmetric Fabry-Perot cavity, the authors demonstrate abrupt phase change at the (nearly) zero reflection point for a specific angle of incidence and wavelength. By exploiting sensor's concept, it was achieved mass sensitivity for relatively heavier molecules such as streptavidin at a level of 1pM concentration due to abrupt change of phase at the reflectionless point. The authors show a nice agreement between the experiment and the theory. The material is well structured and clearly written. The measurements are well made. The paper is of interest to the photonics community, especially to the researchers interested in biological sensing.*

Author Reply: We would like to thank the reviewer for the time and efforts dedicated in reviewing the manuscript and providing valuable feedback and comments. We have clarified and addressed all the critical points raised by the reviewer and carried out additional experiments to validate our claims. Here, we have provided detailed answers to the comments and used the feedback to revise the manuscript accordingly. We fabricated new set of samples and performed detailed experimental analysis that has now addressed the major concerns. We believe that the overall quality of the paper is now improved.

Comment 1: *The authors claim in the introduction and final part of manuscript that "this could make its application to ultrathin layer of biotin-thiol to capture streptavidin". It is not immediately clear how and why this would work, taking into consideration how biological*

molecules would bind to the sample, especially to thin top Ag layer (18 nm) which is not the best for bio-functionalization and suffers the chemical instability due to the oxidation with time.

Author Reply: We thank the reviewer for this important question. To immobilize thiolated-biotin on the sensor surface we have introduced a new chemistry to first prepare thiol-functionalized biotin. We have described the details of synthetic procedure under the section ‘Methods (Surface Chemistry)’. We have utilized thiophilic feature of silver surface to bound thiolated-biotin on the surface. The synthetic scheme for the preparation of thiol-functionalized biotin molecule is shown below:

Scheme showing preparation of biotin disulphide (1):

Scheme showing preparation of biotin thiol (2):

In the next step, we have conducted X-ray photoelectron spectroscopy (XPS) to confirm the effective functionalization of biotin-thiol utilizing Ag-S linkage. Figure R1a shows XPS survey spectrum of bare sample (top Ag layer). High-resolution XPS analysis and the peak at

368.2 eV region indicates metallic Ag on the surface (Figure S2b). Measurements were carried out at multiple regions to confirm the observation and presence of no organic traces were recorded from bare sample.

Fig. R1: XPS spectrum of Ag surface without any functionalization. a) survey, b) peaks corresponding to metallic Ag (Ag 3d).

Fig. R2: XPS spectrum of Ag surface after modification with biotin-thiol. a) survey, peaks corresponding to region of b) metallic Ag (Ag 3d), c) C 1s, d) N 1s, e) O 1s and f) S.

The sensor surface (top Ag layer) was then modified using biotin-thiol (as described in the Methods section) and ensure complete incubation on the surface. The excess biotin-thiol were washed thoroughly using PBS buffer and the samples were analyzed immediately after 12 h freeze-drying process. Figure R2a shows XPS survey spectrum of thiolated biotin containing Ag. Figure R2b indicates a shift in the binding energy to 368 eV, which indicates the effective surface change due to the attachment of electron donating ligand (here thiol functional group from biotin-thiol). Similarly, XPS analysis further revealed peaks in the region corresponding to C 1s (Figure R2c), N 1s (Figure R2d), O 1s (Figure R2e) and S (Figure R2f). The presence of elements C, N, O and S on the surface further confirms the functionalization of the surface using biotin-thiol.

This section has been included in the supplementary file (as Section 3) and following text has been added in the manuscript (Page 11),

“Surface functionalization was further confirmed using X-ray photoelectron spectroscopy analysis by tracing Sulphur and Nitrogen residues post-modification (see Supplementary Section 3).”

We have also performed biosensing experiments by functionalizing the Ag surface with different concentrations of biotin (0.1 mM, 1 mM and 10 mM). We observed that ψ_{min} values increased with increasing biotin concentration. It shows that biotin molecules are strongly attached to Ag surface. In addition, ψ_{min} increased after exposing 1pM streptavidin concentration, which indicates the clear binding of streptavidin molecules on the biotin functionalized sensor surface. The biosensing results obtained using 0.1 mM, 1 mM and 10 mM biotin attached samples are shown in Fig. R3, Fig. R4 and Fig. R5, respectively.

Fig. R3 Biosensing data for 0.1 mM biotin attached sample. Ellipsometry parameters (a) ψ and (b) Δ versus wavelength before (black curve) and after (red curve) exposing 1pM streptavidin (at 73°). Ellipsometry parameters (c) ψ and (d) Δ versus incident angle before (black curve) and after (red curve) exposing 1pM streptavidin (at 730 nm).

Fig. R4 Biosensing data for 1 mM biotin attached sample. Ellipsometry parameters (a) ψ and (b) Δ versus wavelength before (black curve) and after (red curve) exposing 1 pM streptavidin (at 65°). Ellipsometry parameters (c) ψ and (d) Δ versus incident angle before (black curve) and after (red curve) exposing 1 pM streptavidin (at 708 nm).

Fig. R5 Biosensing data for 10 mM biotin attached sample. Ellipsometry parameters (a) ψ and (b) Δ versus wavelength before (black curve) and after (red curve) exposing 1pM streptavidin (at 75°). Ellipsometry parameters (c) ψ and (d) Δ versus incident angle before (black curve) and after (red curve) exposing 1pM streptavidin (at 696 nm).

This section has been included in the supplementary file as Section 4

To check the oxidation issue of top Ag surface, we have recorded the ellipsometry spectra of bare sample two times, two days and 30 days after fabrication. The ellipsometry spectra of bare sample recorded for both times are shown in Fig. R6. As can be seen, no considerable change in resonance wavelength and incident angles are obtained after 30 days. However, ψ_{min} is slightly increased to 0.401° after 30 days (first time it was 0.26° in wavelength scan), which could be due to oxidation of thin Ag surface. Note that this slight increase of ψ_{min}

not at all affect the biosensing performance because close to zero reflection condition ($\psi_{min}=1^\circ$ to 3°) is sufficient for biosensing. We observe that ψ_{min} slightly increases after functionalization of the surface with biotin (see Fig. R3, Fig.R4 & Fig. R5), therefore the oxidation of top Ag layer is not very critical here. We have also recorded the XPS spectra both times and did not identify any peak related to oxygen. It further confirms that oxidation does not affect the functionalization of biotin on the sensor surface.

Fig. R6. Twice recorded ψ and reflection (*p*- and *s*-polarization) spectra at 61° angle of incidence stretched over a period of 2 and 30 days, respectively.

In addition, silver nanoparticles and silver thin films were widely used for plasmonic biosensing applications (*Anal. Chem.* 82, 6350 (2010), *J. Phy. Chem. B* 128, 2115 (2006), *Analytica Chimica Acta* 751, 24-43 (2012), *ACS Applied Materials and Interfaces* 5, 10853-7 (2013), *Current Opinion in Chemical Engineering* 1, 3-10 (2011)). This oxidation issue can be overcome by using capping layers such as platinum or gold (3-5 nm thickness). Alternatively, optimized thickness of bimetallic silver/gold layers (gold as an outer layer) or silver/ platinum layers (platinum as an outer layer) can be used (*Zynio et al., Sensors* 2, 62-70 (2002)). In addition, top silver layer can be replaced by platinum or sputtered gold layer. In order to study

the top metal film effect on the ellipsometry parameters, we have replaced top Ag film with thermally evaporated gold film and recorded the data for different thickness of gold layer. The lowest ψ_{min} is obtained for an optimized top Au layer thickness of 19 nm. The ellipsometry parameters obtained at 65° angle of incidence are shown in Fig. R7. As shown in Fig. R7b, a lowest ψ_{min} of 4.1° and a small phase change is obtained for the optimized geometry. The experimental and simulated reflection spectra for both p - and s -polarizations are shown in Fig. R7c & R7d. Both experimental spectra are in good agreement with simulation spectra. It shows that selection of top metal layer is crucial to achieve $\psi_{min} < 1$ and maximum phase change.

Fig. R7. Experimentally determined ellipsometry parameters (ψ and Δ) and reflection spectra for samples with top metal layer as gold. (a) Wavelength scan spectra recorded at 65° (b) Angular scan spectra recorded at 718 nm, (c) experimental p - and s -polarized reflection spectra at 65° and (d) simulated p - and s -polarized reflection spectra at 65° .

Since silver thin film is low loss compared gold thin film, silver is the best metal to realize point of darkness and extreme phase singularity. It shows that thermally evaporated gold film is not a good alternate for top Ag layer. Therefore, the potential alternative metal could be platinum or sputtered gold.

This section has been included in the supplementary file (Section 6).

The following text has been added in the discussion section of the manuscript,

“Since top layer is silver thin film, the oxidation of silver film could slightly increase ψ_{min} (see Supplementary Fig.19). The oxidation issue can be resolved to a large extent by using capping layers such as platinum or gold (3-5 nm thickness), or optimized thickness of bimetallic silver/gold layers (gold as an outer layer)⁴⁶. (Page 13)

Comment 2: *In the manuscript, the authors explain the relevance of their work in the context of phase sensitive a simple metal-dielectric multilayer thin film stack. The so-called "topological darkness," is a known property of classical electromagnetism. It is the generalization of the Brewster's angle in the complex plane and multi-layered systems and has been observed many times in the past. Could authors suggest the way (recipe) to achieve the supper narrow resonances for such simple layered structures?*

Author Reply: We do agree that there are some experimental and theoretical works, which discuss about the perfect absorption of light using multi-layered systems. However, to our best of knowledge, *the complete suppression of light ($R \approx 0$) and the phase singularity at the point of darkness using optically thin multi-layered structure is not yet realized.* Importantly, we have experimentally achieved lowest psi value, $\psi_{min}=0.13^\circ$ ($R=0.001\% \approx 0$) and a large phase change of 122° using an optically thin metal-dielectric multilayer film. Therefore, the obtained figure-of-

merit (FOM) of the proposed lithography-free system at optical wavelength (814 nm) is 445, which is 2.2 times higher than the previously reported FOM (FOM of 200 [where $\psi_{min}=0.3^\circ$] of plasmonic gold nanoarrays as reported in *Nat. Mater.* **12**, 304-309 (2013)). We have further exploited this phase singularity property at the point of darkness for the phase-sensitive sensing of biomolecules at lower concentrations, which has not been demonstrated so far using thin metal-dielectric film stacks.

In short, the novelty of our work is the following:

- This is the first work which experimentally demonstrates the lowest ψ_{min} (0.13°) and record FOM (445) using *lithography-free* geometry. Previously reported FOM was 200, which was realized using nanopatterned plasmonic gold nanoarrays [*Nat. Mater.* **12**, 304-309 (2013)].
- This is the first work which demonstrates the *extreme phase singularity* at the point of darkness using *lithography-free* geometry.
- Introduces novel chemistry for thiol functionalization of biotin molecules and its immobilization on Ag-surface.
- First experimental demonstration of phase-sensitive biosensing at lowest concentration using *lithography-free* structures.

To put the present work in the proper perspective, we have amended the respective sentences in the manuscript as,

“Here, we propose and experimentally demonstrate a new approach to realize perfect absorption (99.99%) and extreme phase singularity using a simple metal-dielectric multilayer thin film stack.” (Abstract, Page 2)

“The abrupt phase change at the point of darkness so far has only been demonstrated based on the concept of topological darkness which requires complex lithography techniques for fabricating nanostructures to engineer the effective index. In this work, we have experimentally realized a new route to achieve abrupt phase change at the point of darkness in a lithography-free four layered metal-dielectric system at the Brewster angle.’ (Discussion, Page 12)

Title of the paper has been amended as,

“Realization of singular phase in optically thin metal-dielectric films for ultrasensitive biosensing applications”

We have included the following text and references related to perfect absorption based on thin film structures in the manuscript:

“Even though the perfect absorption has been demonstrated at different spectral regions using various metal-dielectric thin film stacks ³⁵⁻⁴², the extreme singular phase behavior is not yet realized.” (Page 5)

35. Kats, M. A. *et al.* Ultra-thin perfect absorber employing a tunable phase change material. *Appl. Phys. Lett.* **101**, 221101 (2012).
36. Li, Z., Butun, S. & Aydin, K. Large-area, lithography-free super absorbers and color filters at visible frequencies using ultrathin metallic films. *ACS Photonics* **2**, 183-188 (2015).
37. You, J.-B., Lee, W.-J., Won, D & Yu, K. Multiband perfect absorbers using metal-dielectric films with optically dense medium for angle and polarization insensitive operation. *Opt. Exp.* **22**, 8339-8348 (2014).

38. Shu, S., Li, Z. & Li, Y. Y. Triple-layer Fabry-Perot absorber with near-perfect absorption in visible and near-infrared regime. *Opt. Exp.* **21**, 25307-25315 (2013).
39. ElKabbash, M. *et al.* Iridescence-free and narrowband perfect light absorption in critically coupled metal high-index dielectric cavities. *Opt. Lett.* **42**, 3598-3601 (2017).
40. Kats, M. A. & Capasso, F. Optical absorbers based on strong interference in ultra-thin films. *Laser & Photonics Reviews* **10**, 699 (2016).
41. Park, J. *et al.* Omnidirectional near-unity absorption in an ultrathin planar semiconductor layer on a metal substrate. *ACS Photonics* **1**, 812-821 (2014)
42. Kocer, H.; Butun, S., Li, Z. & Aydin, K. Reduced near infrared absorption using ultra-thin lossy metals in Fabry- Perot cavities. *Sci. Rep.* **5**, 8157 (2015) (Page 19)

As another point raised by the reviewer, “Could authors suggest the way (recipe) to achieve the super narrow resonances for such simple layered structures?” To answer this comment, we have fabricated new set of samples by optimizing the thickness of each layer to achieve minimum ψ and maximum phase change. The optimized thickness of each layer in the proposed geometry is: top Ag layer (20 nm), MMA layer (522 nm), Ge layer (10 nm) and bottom Ag layer (80 nm).

In Fig. R8, we plot the experimental ellipsometry spectra (ψ and Δ) and p -polarized reflection spectra of the samples ‘with’ and ‘without’ Ge layer for different thickness of MMA layer. All the spectra are plotted for an incident angle at which minimum ψ is obtained for the longer wavelength mode. It is clear that ‘without Ge’ samples show narrow resonance without any singular phase behavior (Fig. R8 a, d & g). However, the linewidth of the resonance decreases and singular phase behavior appears when Ge layer is present in the geometry (Fig. R8 b, e & h). It is also evident that the linewidth of the resonance changes by varying MMA layer thickness.

Note that extreme singular phase is obtained for 522 nm thick MMA layer (Fig. R8e), where the linewidth of the resonance is maximum. It shows that it is not possible to achieve extreme singular phase and narrower resonance simultaneously using Ge layer.

Fig. R8. Experimentally determined ellipsometry parameters (ψ and Δ) and p-polarized reflection spectra of samples ‘with’ and ‘without’ Ge layer. (a) to (c) for 630 nm MMA layer, (d) to (f) for 522 nm MMA layer and (g) to (i) for 443 nm MMA layer. The incident angle is fixed where the ψ_{min} is obtained for the longer wavelength mode.

Fig. R9. Experimentally determined ellipsometry parameters (ψ and Δ). (a) using 10 nm GST layer and (b) using 10 nm Ge layer.

Nevertheless, it is possible to achieve narrow resonance and singular phase by replacing Ge layer with another high index dielectric such as $\text{Ge}_2\text{Sb}_2\text{Te}_5$ (GST). Experimentally obtained ellipsometry parameters of the structure using 10 nm GST layer is shown in Fig. R9a. The thickness of other layers used in the structure are: top Ag (20 nm), MMA (522 nm) and bottom Ag (50 nm). As can be seen, here both modes show narrow resonance and singular phase behavior. It shows that singular phase depends on ψ_{\min} value as well as the narrow resonance. In contrast to Ge layer sample (Fig. R9b), extreme singularity is not achieved for both modes because the obtained $\psi_{\min} > 1$. However, we still believe that it is possible to achieve super narrow resonance and extreme phase singularity simultaneously by optimizing the thickness of MMA layer, GST layer and Ag layer. In addition, a tunable point of darkness condition and singular phase is possible for this geometry, by switching the phase of the GST from amorphous to crystalline. Note that both narrow resonance with high quality factor and extreme singular phase are equally important for sensing applications. We are currently working in this direction and the results will be reported elsewhere.

This section has been included in the supplementary file (Section 2 and Section 5)

The following text has been included in the manuscript (Page 12),

“In contrast to plasmonic gold nanoarray-biosensor based on singular phase concept¹¹, the device sensitivity of the proposed sensor could be further improved. Since the higher phase sensitivity of plasmonic gold nanoarray is due to super narrow mode plasmonic resonance¹¹, it could be possible to improve the device sensitivity of proposed sensor further by narrowing the linewidth of the cavity mode (by achieving high-quality factor mode). In fact, we have achieved narrow linewidth mode by replacing Ge layer with phase change material such as $\text{Ge}_2\text{Sb}_2\text{Te}_5$ (see Supplementary Fig. 18).”

Comment 3: *Usually thin metallic and semiconductor films are not continuous below ~10-15 nm under thermal evaporation conditions. There will be islands on the surface of film, which can be seen in high resolution SEM or TEM. Could authors provide structural characteristics for Ge film; this will impact the validity of work.*

Author Reply: We have carried out Field Emission Scanning Electron Microscopy (FE-SEM) to understand the surface irregularities under thermal evaporation conditions. FE-SEM image of Ge surface is shown in Fig. R10, which indicates that thermally evaporated 10 nm thick Ge film has smooth surface texture and morphology.

Fig. R10: FE-SEM images of thermally evaporated 10 nm thick Ge film

FE-SEM image has been included in the supplementary file (as Fig. S2) and following text has been added in the manuscript (Page 6)

“Field Emission Scanning Electron Microscopy (FE-SEM) images of Ge surface is shown in Supplementary Fig. 2, which indicates that thermally evaporated 10 nm thick Ge film has uniform surface texture and morphology”.

Comment 4: *It is mentioned that data is acquired from 400nm-1000nm. Nevertheless, the data shown is only between 700 and 1000nm (Fig.3) for spectral dependence of reflections, while the data (Fig. 1) for ellipsometric functions is shown from 400 to 1000nm. I suggest being consistent and using the same wavelength scale in both figures for clarity.*

Author Reply: As suggested by the reviewer, we have used same wavelength scan range (450-1000 nm) in both Fig. 1 and Fig.3. The new Figure 3 obtained using the optimized geometry is shown in Fig. R11.

Fig. R11 Reflectivity spectra and field distribution. (a) Experimental and (b) calculated reflection spectra for p- and s-polarization at 61° angle of incidence. (c) Calculated field distribution along the four-layered system at the point of darkness condition (814 nm and 61°) and (d) its magnitude as a function of depth along the multilayer.

Comment 5: Authors should provide the detailed and deep explanation for Figs. 1d and. 3a: from 800-950 nm shown in Fig. 3(a) and (b), why the absolute value in R_p is zero, but for P_{si} approaches are non-zero? Although the data is very compelling, P_{si} indeed approaches close to zero, the data does not support P_{si} reaching absolute zero in the experiment.

Author Reply: In Fig. 1d, we have presented the results of proposed sample without top Ag layer. This sample shows totally different spectra compared to other three samples (Fig. 1a to 1c). First of all, we want to clarify that no phase change is observed for this sample.

Interestingly, completely reversed ψ spectrum is obtained. That means, a reflection peak instead of reflection dip is observed in the ψ spectrum. In order to clearly understand this, we have recorded the ellipsometry parameters (ψ and Δ) and p - and s -polarized reflection spectrum at higher angles of incidence (60° to 80°). In Fig. R12, we present the results of this sample for different angles of incidence. As shown in Fig. R12 a to c, the cavity modes are converted into reflection peaks in ψ spectra. This is because the s -polarized light provides minimum reflection at that spectral wavelength as compared p -polarized light, which is shown in Fig. R12 d to f. However, for other three samples (Fig. 1a, b & c), p -polarized light provided the minimum reflection at the resonance wavelengths as compared to s -polarized light, so that reflection dip was obtained. A blue shift in resonance wavelength is obtained with increasing angle of incidence, which is similar to the case with other three samples. Also, at a particular incident angle (70°) higher quality factor mode with maximum intensity is obtained at shorter wavelength, which is around 495 nm. Interestingly, s -polarized reflection is almost zero at that wavelength (Fig. R12e). That is why a sharp reflection peak is obtained at 495 nm wavelength. In short, the proposed sample (Fig. 1c) provides zero reflection for p -polarized light at a particular angle of incidence for longer wavelength mode so that lowest ψ is possible for that mode. However, the sample without top Ag layer (Fig. 1c) provides zero reflection for s -polarized light at a particular angle of incidence for the shorter wavelength mode so that highest ψ is possible for that mode. This behavior is because $\tan \Psi = \left| \frac{r_p}{r_s} \right|$. The physical mechanism of achieving almost zero reflection for s -polarized light using the proposed sample without top Ag layer could be due to generalized Brewster angle (pseudo) effect (Mahlein, H. F. *JOSA* 64, 647-653 (1974)). We further calculated p - and s -polarized reflection spectra of this sample for

different angles of incidence using TMM (Fig. R16 g to i), which are in very good agreement with experimental reflection spectra.

Fig. R12 Ellipsometry parameters (ψ and Δ) and p - and s -polarized reflection spectra of proposed sample without top Ag layer for different angles of incidence. (a)-(c) Ellipsometry parameters. (d)- (f) Experimental reflection spectra for p - and s -polarization and (g)-(i) Simulated reflection spectra for p - and s -polarization.

This section has been included in the supplementary file (Section 2) and following text has been included in the manuscript,

“The ψ spectrum of the system presented in Fig. 1d shows a reverse behavior owing to the fact that the s -polarized light provides a minimum reflection at the resonance wavelength when compared to the p -polarized light (see Supplementary Fig. 8)” (Page 7)

Fig. R13. Experimental angular and polarization depended absorption spectra of ‘without Ge’ and ‘with Ge’ samples. Absorption spectra of ‘without Ge’ sample (a) p-polarization, and (b) s-polarization. . Absorption spectra of ‘with Ge’ sample (c) p-polarization, and (d) s-polarization.

Experimentally recorded reflection spectra of the proposed geometry for *p*- and *s*-polarization are shown in Fig. 3a. Actually, we have obtained reduced reflection for *p*-polarization and increased reflection for *s*-polarization at the point of darkness region after introducing Ge layer in the sample. To explain this behavior, we first present a detailed explanation for this interesting phase singularity behavior of the proposed sample by comparing the results of two samples (‘with’ and ‘without’ Ge layer). In particular, we have experimentally investigated the influence of incident angle and polarization states on the absorption ($Abs=1-R$,

where $T=0$) spectra of ‘without Ge’ and ‘with Ge’ samples. The absorption spectra as a function of incident angle (20° to 80°) and polarization states (p - and s -polarization) are shown in Fig. R13. In the case of ‘without Ge’ sample, for p -polarization (Fig. R13a), almost perfect absorption is obtained for all the angles of incidence and narrow band perfect absorption is obtained for s -polarization (Fig. R13b) at lower incident angles (below 35°). For ‘with Ge’ sample (Fig. R13c), a wide band perfect absorption for longer wavelength mode at higher incident angle is obtained for p -polarization. However, no perfect absorption is observed for s -polarization (Fig. R13d) even at lower incident angles. In particular, p -polarized absorption increased (Fig. R14a) and s -polarized absorption decreased (Fig. R14b) for ‘with Ge’ sample as compared to ‘without Ge’ sample. The p - and s -polarized absorption spectra recorded at ψ_{min} angle of ‘with Ge’ and ‘without Ge’ samples are shown in Fig. R14. Since ψ_{min} is obtained at 814 nm for ‘with Ge’ sample, the measured p - and s -polarized absorption at 814 nm wavelength are 99.999% and 63 %, respectively. On the other hand, p -polarized reflection decreased and s -polarized reflection increased at ψ_{min} point after introducing a thin Ge layer in the sample.

Fig. R14. Absorption spectra of ‘with Ge’ and ‘without Ge’ samples at ψ_{min} angle. (a) for p -polarization and (b) for s -polarization.

This is the reason for ‘with Ge’ samples to show reduced ψ_{min} and sharp singularity as compared to ‘without Ge’ samples because $\tan \Psi = \left| \frac{r_p}{r_s} \right|$. However, both p - and s -polarized spectrum broadened after introducing the Ge layer.

As another point raised by the reviewer as to why the absolute value in R_p is zero, but for Ψ approaches non-zero. The answer for this question is given above. It is clear from Fig. R14 that R_p approaches zero and R_s increases at the region of point of darkness. For this reason, Ψ should not reach absolute zero ($\tan \Psi = \left| \frac{r_p}{r_s} \right|$).

As the last comment by reviewer, “Although the data is very compelling, Ψ indeed approaches close to zero, the data does not support Ψ reaching absolute zero in the experiment”. As mentioned above, Ψ should not reach absolute zero. In our experiments and simulation, we did not obtain absolute zero for Ψ . The minimum Ψ obtained in experiment is at 0.13° (in angular scan) and then we have tried to reproduce the same value in our simulation by optimizing the simulation parameters. The value of Ψ also depends on wavelength and angle scanning step.

This section has been included in the supplementary file (Section 4).

Comment 6: *Measured ellipsometric parameters Ψ and Δ for biotin-thiol covered sample are impressionable because of the ellipsometer functions are rather physical. Meaningful would be the p -polarized reflection spectra for those findings, rather than Ψ .*

Author Reply: As suggested by the reviewer, we have recorded the p -polarized reflection for biotin-thiol covered samples, in addition to Ψ and Δ spectrum. Note that ψ_{min} is obtained at higher incident angles and the corresponding resonance wavelengths were blue shifted, after

functionalizing the bare samples with different concentrations of biotin. This is because the surface roughness of the sample was slightly increased due to the extra thin layer of biotin-thiol. The thickness of this layer slightly increases as the concentration increases.

Fig. R15 Biosensing characterization using p-polarized reflection spectrum: Measured R_p spectrum of the samples (a) 0.1 mM biotin covered at 73° , (b) 1 mM biotin covered at 65° and (c) 10 mM biotin covered at 75° .

As shown in Fig. R15a, reflection close to zero is obtained for 0.1 mM biotin covered sample because 0.1 mM biotin only add an extra layer thickness <1 nm on the bare sample. It should be noted that there is not much difference in the spectral sensitivity when the concentration of functionalized molecules is changed. The maximum wavelength shift (2 nm) is obtained for 1 mM biotin covered sample. However, the phase sensitivity is better for 0.1 mM biotin covered sample because of the lowest ψ_{min} value. The recorded maximum phase sensitivity after exposing 1 pM streptavidin on 0.1 mM and 1 mM biotin covered sample is 26° and 33° , respectively (see Fig. R3 & Fig. R4). Note that 1 mM biotin sample provided slightly higher phase sensitivity due to the availability of large number of biotin binding sites on the sample. Since it is well known in Streptavidin-biotin system that four biotin molecules bind to one streptavidin molecule, more biotin molecules are available for streptavidin to bind in 1 mM biotin sample as compared to 0.1 mM biotin sample. The observed ψ change ($\delta\psi=5.9^\circ$) also confirms that specific binding is strong in the case of 1 mM biotin. However, in the case of 10

mM biotin covered sample weak sensitivity is obtained because ψ_{min} value is largely enhanced to 4.1° (Fig. R5). In addition, as the concentration increases, the possibility of multiple adsorbed molecules leads to interference effects, with each additional molecule having a decreasing impact on the phase shift.

The p -polarized reflection spectrum also included in the spectrum of Psi and Delta in the manuscript (Fig. 4) and supplementary file (Fig. S14 to S17)

The following text has been included in the revised manuscript (Page 11),

“The p -polarized reflection spectra also shown in the inset of Fig. 4b.

Comment 7: *In Fig. S1 authors display the measured optical constants (n and k), using Woollams' ellipsometry, of thin films: Ag, Ge, MMA and Si. These data were taken for thick film Ag(50nm), Ge (100nm) but in the four-layered system in the case of darkness condition the thickness of corresponding film is smaller: Ag (30nm), Ge(10nm). It is well known that the complex refractive index of thermally deposited films strongly depends on thickness of film (for small thickness $<50\text{nm}$) and conditions of growing. It is not clear why authors did not measure the films with real thicknesses of metal-dielectric multilayer thin film stack. Another question about data at Fig. S1, why for Ge film authors present the real and imaginary parts of the refractive index but for Si film only the real part, n ?*

Author Reply: We thank the referee for this comment. We do agree that the complex refractive index of thermally deposited thin films strongly depend on thickness of film and growth conditions. As suggested by the reviewer, we have determined the complex refractive index of exact thickness of Ag and Ge used in the proposed optimized geometry, which are 20 nm for top Ag, 10 nm for Ge and 80 nm for bottom Ag. In Fig. R16, we present the ellipsometrically

determined and fitted optical constants of Ag (20 nm), Ge (10 nm), and MMA (522 nm). However, the complex refractive index of substrate (Si) was directly taken from the existing model of V-VASE software.

Fig. R16 Measured optical constants (n & k) of thin films. (a) Ag (20 nm), (b) Ge (10 nm), (c) MMA, and (d) Si.

In the previous version, we did not include the imaginary parts of Si because we have only considered real parts of Si refractive in the calculations. In the revised version, we have used both real and imaginary parts of Si refractive index in the simulations. In all our calculations, the optical constants obtained using the exact thickness of thin films were used.

The growing conditions of Ag and Ge thin films are included in the Methods section and Fig. S1 in the supplementary file is replaced with new figure (Fig. R16).

Reply to Reviewer-2

Reviewer#2

This is a well written manuscript which reports advances in ultra-sensitive biodetection on a multilayer structure without nanofabrication.

Author Reply: We would like to thank the reviewer for the time and efforts dedicated in reviewing the manuscript and for providing valuable feedback and comments which helped us tremendously to sharpen the manuscript. We have clarified and addressed all the critical points raised by the reviewer. Here, we provide detailed answers to the comments and used the feedback to revise the manuscript accordingly. We have fabricated new set of samples and performed detailed experimental analysis which address the major concerns related to the novelty and biomolecule binding of the work. We now believe that the overall quality of the paper has significantly improved.

Comment 1: Novelty: *As the authors point out, multilayered metal-dielectric perfect absorbers similar to the ones reported here and that are not directly related to topological darkness and do not require nanopatterning have been studied in the literature before, in theory and in experiment, recent examples are DOI:10.1364/OE.22.008339 | OPTICS EXPRESS 8339 (theory + exp); S. Shu, Z. Li, and Y. Y. Li, "Triple-layer Fabry-Perot absorber with near-perfect absorption in visible and nearinfrared regime," Opt. Express 21(21), 25307–25315 (2013). The impact of their work therefore rests on the application of these structures in biosensing. Here, I would question if pM detection of Streptavidin is a breakthrough that warrants publication in a Nature Journal.*

Author Reply: We sincerely apologize for being ignorant and missing out the relevant important works in the literature. We do agree that there are some experimental (*APL* 101, 221101(2012) and *ACS Photonics* 2, 183(2015)) and theoretical (*Opt. Exp.* 21, 25307 (2013) and *Opt. Exp.* 22, 8339 (2014)) works which discuss about the perfect absorption of light using lithography-free thin film multilayer stacks. However, *to our best of knowledge, the complete suppression of light ($R \approx 0$) and the phase singularity at the point of darkness using optically thin multilayer structure is not yet realized.* Importantly, we have experimentally achieved lowest psi value, $\psi_{min}=0.13^\circ$ ($R=0.001\% \approx 0$) and a large phase change of 122° using an optically thin metal-dielectric multilayer film. Therefore, the obtained figure-of-merit (FOM) of the proposed *lithography-free* system at optical wavelength (814 nm) is 445, which is 2.2 times higher than the previously reported FOM (FOM of 200 [where $\psi_{min}=0.3^\circ$] has been realized using *nanopatterned* plasmonic gold nanoarrays, *Nat. Mater.* 12, 304-309 (2013)). We have further exploited this phase singularity property at the point of darkness for the phase-sensitive sensing of biomolecules at lower concentrations, which is not demonstrated so far using thin metal-dielectric film stacks.

As the reviewer pointed out, the two published papers (*Opt. Exp.* 21, 25307 (2013) and *Opt. Exp.* 22, 8339 (2014)) have indeed reported perfect absorption using thin film structures. The authors of *Opt. Exp.* 21, 25307 (2013) reported the near-perfect absorption of light in the visible and near-infrared regions using an unpatterned metal/dielectric/metal triple-layer. In *Opt. Exp.* 22, 8339 (2014), the cavity resonant properties of planar metal-dielectric layered structures with optically dense dielectric media were studied with the aim of realizing omnidirectional and polarization-insensitive operation. *It should be noted that both are elegant theoretical works and no phase singularity was reported in either of the works.* However, in the present work, we experimentally report the multiband and wide angle perfect absorption, with a maximum of

99.999% at 814 nm wavelength (See Fig. R4). Importantly, we show an abrupt phase change at the maximum absorption (minimum reflection) point, which is extremely useful for ultrasensitive biosensing applications. It should be noted that our proposed geometry ‘without Ge’ layer even provides wide angle perfect absorption with a maximum of **99.86%** at 830 nm (See Fig. R4), but phase singularity behavior disappears due to the absence of an absorbing dielectric layer. It shows that phase singularity is only possible when the multilayer film consists of a very thin layer of highly absorbing dielectric such as Ge. Therefore, we would like to clarify that the impact of this work not only rests on the application of the F-P cavity in biosensing but also the proposed lithography-free geometry being a good candidate to achieve extreme phase singularity (previously it was realized only using nanopatterned geometries). In the following sections, we present a detailed explanation for this interesting phase singularity behavior of the proposed sample by comparing the results of various samples (‘with’ and ‘without’ Ge layer).

We have fabricated new set of samples by optimizing the thickness of each layer to achieve minimum ψ and maximum phase change. The optimized thickness of each layer in the proposed geometry is: top Ag layer (20 nm), second MMA layer (522 nm), Ge layer (10 nm) and bottom Ag layer (80 nm). Initially, we have optimized the thickness of cavity (MMA) layer to achieve minimum ψ and the maximum phase change. For this purpose, we have first determined the thickness of MMA layer by spin coating the MMA on Si wafer at different r. p. m. Ellipsometrically determined thickness of MMA layer at 2000, 3000 and 4000 r. p. m. are \square 630 nm, \square 522 nm and \square 443 nm, respectively. Thickness of other layers in the multilayer are: top Ag layer (20 nm), Ge layer (10 nm) and bottom Ag layer (80 nm). In Fig. R1, we plot the experimental ellipsometry spectra (ψ and Δ) and p -polarized reflection spectra of the samples ‘with’ and ‘without’ Ge layer for different MMA layer thickness. All the spectra were plotted for

an incident angle at which minimum ψ is obtained for the longer wavelength mode. Importantly, a singular phase behavior at ψ_{min} point and a blue shift in resonance is obtained for all the thickness of MMA layer by introducing a thin Ge layer (10 nm) in the geometry. It shows that phase singularity is only possible when Ge layer is present in the geometry. One can see that the modes are blue shifted by decreasing the thickness of MMA layer and the lowest ψ value, and minimum p -polarized reflection are obtained for a MMA layer of thickness 522 nm (Figs. d to f).

Fig. R1. Experimentally determined ellipsometry parameters (ψ and Δ) and p -polarized reflection spectra of samples ‘with’ and ‘without’ Ge layer. (a) to (c) for 630 nm MMA layer, (d) to (f) for 522 nm MMA layer and (g) to (i) for 443 nm MMA layer. The incident angle is fixed where the ψ_{min} is obtained for the longer wavelength mode.

Fig. R2. Measured pair of ellipsometry parameters as a function of incident angle. (a) for ‘without Ge’ sample at 835 nm and (b) for ‘with Ge’ sample at 814 nm.

For 522 nm thick MMA layer, the obtained ψ_{min} for the longer wavelength mode (above 800 nm) is 3° ($R=0.008$ at 835 nm) for ‘without Ge’ sample and 0.26° ($R= 0.0000114$ at 814 nm) for ‘with Ge’ sample. In contrast to ‘without Ge’ sample, ψ_{min} is reduced 11 times (reflection reduced to 725 times) and a sharp change in phase is obtained by using ‘with Ge’ sample. It should be noted that reduction in ψ_{min} and higher phase change at ψ_{min} is possible for angular scan as compared to wavelength scan. Therefore, in Fig. R2, we plot the ellipsometry parameters of the samples ‘without’ and ‘with’ Ge layer, as a function of incident angle. One can see that ψ_{min} is exactly obtained at the minimum incident angle for both samples (55° for ‘without Ge’ and 61° for ‘with Ge’). It is evident from the results that ‘with Ge’ sample only shows an abrupt phase change at ψ_{min} point (obtained 122° phase change). The obtained ψ_{min} using angular scan is 0.13° (23 times reduction compared to ‘without Ge’ sample) and the calculated FOM ($1/\psi_{min}$, where ψ is expressed in radians) is 445, which is the highest value reported so far based on singular phase concept. It indicates that a thin layer of Ge plays an important role in the geometry to achieve point of darkness and phase singularity.

Fig. R3. Experimental angular and polarization depended absorption spectra of ‘without Ge’ and ‘with Ge’ samples. Absorption spectra of ‘without Ge’ sample (a) *p*-polarization, and (b) *s*-polarization. . Absorption spectra of ‘with Ge’ sample (c) *p*-polarization, and (d) *s*-polarization.

We then experimentally investigated the influence of incident angle and polarization states on absorption ($Abs=1-R$, where $T=0$) spectra of ‘without Ge’ and ‘with Ge’ samples. The absorption spectra as a function of incident angle (20° to 80°) and polarization states (*p*- and *s*-polarization) are shown in Fig. R3. In the case of ‘without Ge’ sample, for *p*-polarization (Fig. R3a), almost perfect absorption is obtained for all the angles of incidence and narrow band perfect absorption is obtained for *s*-polarization (Fig. R3b) at lower incident angles (below 35°). For ‘with Ge’ sample (Fig. R3c), a wide band perfect absorption for longer wavelength mode at

higher incident angle is obtained for p -polarization. However, no perfect absorption is observed for s -polarization (Fig. R3d) even at lower incident angles. In particular, p -polarized absorption increased (Fig. R4a) and s -polarized absorption decreased (Fig. R4b) for ‘with Ge’ sample as compared to ‘without Ge’ sample. The p - and s -polarized absorption spectra recorded at ψ_{min} angle of ‘with Ge’ and ‘without Ge’ samples are shown in Fig. R4. Since ψ_{min} is obtained at 814 nm for ‘with Ge’ sample, the measured p - and s -polarized absorption at 814 nm wavelength are 99.999% and 63%, respectively. On the other hand, p -polarized reflection decreased and s -polarized reflection increased after introducing a thin Ge layer in the sample.

Fig. R4. Absorption spectra of ‘with Ge’ and ‘without Ge’ samples at ψ_{min} angle. (a) for p -polarization and (b) for s -polarization.

This is the reason for ‘with Ge’ samples to show reduced ψ_{min} and sharp singularity as

compared to ‘without Ge’ samples because $\tan \Psi = \left| \frac{r_p}{r_s} \right|$. We have further investigated the Ge

layer thickness on achieving phase singularity at ψ_{min} . In Fig. R5, we present the wavelength and angular scan results of ellipsometry parameters by slightly varying the thickness (8 nm-12 nm) of Ge layer. As can be seen, lowest ψ_{min} and maximum phase change is obtained for 10 nm Ge

thickness. We noticed that phase change is only significant when the Ge thickness is within 8-12 nm. We have also investigated the phase singularity variation by changing the thickness of top and bottom Ag film thickness. It should be noted that the top Ag film thickness should be within 18- 21 nm range to obtain significant phase singularity. However, phase singularity is observed for different thickness of bottom Ag film (20 nm to 100 nm) and the maximum phase change is obtained for 80 nm thick bottom Ag film (Fig. R6).

Fig. R5. Measured ψ and Δ spectra for different thickness of Ge layer. For wavelength scan (a) 8 nm, (b) 10 nm, (c) 12 nm and for angular scan (d) 8 nm, (e) 10 nm, (f) 12 nm . The corresponding incident angle and excitation wavelength is shown in each figure.

It shows that the thickness of Ag layer, MMA layer and Ge layer are crucial for realization lowest ψ_{min} and extreme phase singularity at the point of darkness. Based on our experimental analysis, the optimized thickness required to achieve lowest ψ_{min} and extreme phase singularity are: top Ag (20 nm), MMA (522 nm), Ge (10 nm) and bottom Ag (80 nm).

Fig. R6. Measured ψ and Δ spectra for different thickness of bottom Ag layer. For wavelength scan (a) 30 nm, (b) 50 nm, (c) 80 nm and for angular scan (d) 30 nm, (e) 50 nm, (f) 80 nm . The corresponding incident angle and excitation wavelength is shown in each figure.

This section has been included in the supplementary file (Section 2).

In short, the novelty of our work could be highlighted based on the following points:

- This is the first work which experimentally demonstrates the lowest ψ_{min} (0.13°) and record FOM (445) using *lithography-free* geometry. Previously reported FOM was 200, which was realized using nanopatterned plasmonic gold nanoarrays [*Nat. Mater.* **12**, 304-309 (2013)].
- This is the first work which demonstrates *extreme phase singularity* at the point of darkness using *lithography-free* geometry.
- Introduces novel chemistry for thiol functionalization of biotin molecules and its immobilization on Ag-surface.
- First experimental demonstration of *phase-sensitive biosensing* at lowest concentration using lithography-free structures.

To put the present work in the proper perspective, we have amended the respective sentences in the manuscript as,

“Here, we propose and experimentally demonstrate a new approach to realize perfect absorption (99.99%) and extreme phase singularity using a simple metal-dielectric multilayer thin film stack.” (Abstract, Page 2)

“The abrupt phase change at the point of darkness so far has only been demonstrated based on the concept of topological darkness which requires complex lithography techniques for fabricating nanostructures to engineer the effective index. In this work, we have experimentally realized a new route to achieve abrupt phase change at the point of darkness in a lithography-free four layered metal-dielectric system at the Brewster angle.’ (Discussion, Page 12)

Title of the paper has been amended as,

“Realization of singular phase in optically thin metal-dielectric films for ultrasensitive biosensing applications”

As suggested by the reviewer, we have included the following text and references related to perfect absorption based on thin film structures in the manuscript:

“Even though the perfect absorption has been demonstrated at different spectral regions using various metal-dielectric thin film stacks ³⁵⁻⁴², the extreme singular phase behavior is not yet realized.” (Page 5)

35. Kats, M. A. *et al.* Ultra-thin perfect absorber employing a tunable phase change material. *Appl. Phys. Lett.* **101**, 221101 (2012).

36. Li, Z., Butun, S. & Aydin, K. Large-area, lithography-free super absorbers and color filters at visible frequencies using ultrathin metallic films. *ACS Photonics* **2**, 183-188 (2015).
37. You, J.-B., Lee, W.-J., Won, D & Yu, K. Multiband perfect absorbers using metal-dielectric films with optically dense medium for angle and polarization insensitive operation. *Opt. Exp.* **22**, 8339-8348 (2014).
38. Shu, S., Li, Z. & Li, Y. Y. Triple-layer Fabry-Perot absorber with near-perfect absorption in visible and near-infrared regime. *Opt. Exp.* **21**, 25307-25315 (2013).
39. ElKabbash, M. *et al.* Iridescence-free and narrowband perfect light absorption in critically coupled metal high-index dielectric cavities. *Opt. Lett.* **42**, 3598-3601 (2017).
40. Kats, M. A. & Capasso, F. Optical absorbers based on strong interference in ultra-thin films. *Laser & Photonics Reviews* **10**, 699 (2016).
41. Park, J. *et al.* Omnidirectional near-unity absorption in an ultrathin planar semiconductor layer on a metal substrate. *ACS Photonics* **1**, 812-821 (2014)
42. Kocer, H.; Butun, S., Li, Z. & Aydin, K. Reduced near infrared absorption using ultra-thin lossy metals in Fabry- Perot cavities. *Sci. Rep.* **5**, 8157 (2015) (Page 19)

Comment 2: *It is confusing that the authors first introduce the importance of topologically protected darkness for sensing applications but then state that in fact their multilayer system is not related to this phenomenon: “principle of achieving point of darkness in the proposed system is not directly related to topological darkness phenomenon”. Can the authors please clarify whether their results have any relevance for the “topological darkness phenomenon” and to which degree their point of darkness is topologically protected?*

Author Reply: We thank the referee for his/ her comment and apologize for not being clear on this point. We would like to clarify that the physical mechanism of achieving singular phase in the proposed system is solely due to the presence of highly absorbing dielectric (Ge) layer in the geometry. According our detailed experimental analysis (please see reply to comment 1), it is evident that the singular phase is only visible when an optimized Ge thin film is present in the geometry. We do agree that we have introduced topological darkness concept and phase sensitive sensing based on this concept in the abstract and introduction of the paper because the singular phase and phase sensitive biosensing so far have been demonstrated using the concept of topological darkness. In this context, we have clearly mentioned in the abstract of the paper that “Here, we propose and experimentally demonstrate a new approach to realize extreme phase singularity using a simple metal-dielectric multilayer thin film stack.” In particular, we have introduced the topological darkness concept in the paper to differentiate our work with the existing works. Note that the topological darkness concept is only applicable for optical systems which consists of nanopatterns to engineer their effective index. Since our proposed geometry is independent of nanopatterns and is an asymmetric Fabry-Perot cavity with an optically thick MMA layer, it is not straightforward to determine the effective n and k values of the entire geometry to check whether these n and k values follow the Jordan Curve Theorem to satisfy the topological darkness conditions. It is important to note that we did not obtain singular phase using ‘without Ge’ samples even when top Ag film thickness is the same as that of ‘with Ge’ samples. This is a direct evidence that the topology of the surface does not have any effect on the realization of the singular phase in the proposed geometry.

Fig. R7. Experimentally determined ellipsometry parameters (ψ and Δ) and reflection spectra for samples when top metal layer as gold. (a) Wavelength scan spectra recorded at 65° (b) Angular scan spectra recorded at 718 nm , (c) experimental p- and s-polarized reflection spectra at 65° and (d) simulated p- and s-polarized reflection spectra at 65° .

As shown in reply to comment 1, the thickness of individual layer in the structure has an important role to achieve lowest ψ_{min} and the maximum phase jump. At the same time, the top metal material has an important role in realizing lowest ψ_{min} . Based on our experimental analysis, we have confirmed that top Ag film of thickness 18-21 nm can only provide the lowest ψ_{min} (<1) and large phase change. In order to study the top metal film effect on the ellipsometry parameters, we have replaced top Ag film with gold film and recorded the spectra for different thickness of gold layer. The lowest ψ_{min} is obtained for an optimized top Au layer thickness of 19 nm. The ellipsometry parameters obtained at 65° angle of incidence are shown in Fig. R7. As shown in Fig. R7b, a lowest ψ_{min} of 4.1° and a small phase change is obtained for the optimized

geometry when Au as the top metal layer. The experimental and simulated reflection spectra for both *p*- and *s*-polarizations are shown in Fig. R7c and R7d, respectively. Both reflection spectra match very well. It shows that selection of top metal layer is very important to achieve $\psi_{min} < 1$ and the maximum phase change. Since silver thin film is low loss compared gold thin film, silver is the best metal to realize point of darkness and extreme phase singularity.

The results of top gold layer sample have been included in the supplementary file (Section 6)

Comment 3: Data: All sensing experiments should be repeated at least three times. Best would be to plot all three results or alternatively, error bars should be included in the data plots.

Author Reply: As suggested by the reviewer, we have repeated the sensing experiments many times and results obtained using three different samples from the same batch are presented here. Note that we cannot use the same sample for repeated measurements because it is not reusable once the biomolecules were captured. In order to keep lowest ψ_{min} , we have functionalized the sample surface with 0.1 mM biotin-thiol (it can add a very thin layer of thickness approximately 1 nm on the sample surface). The experimental procedure is as follows: we first recorded the ellipsometry parameters (in both wavelength and angular scan) and *p*-polarized reflection spectra of the biotin-thiol functionalized samples. Then, we have exposed 1 pM streptavidin in PBS for 2h and followed by rinsing with PBS to remove non-specifically bound streptavidin molecules using a detachable PDMS channel with dimension 7 x 7 x 2 mm³. This was performed without removing the sample from the ellipsometer stage in order to keep the same incident beam position on the sample so that we could extract the correct shift. Note that we have removed the PDMS channel while doing the measurements because no phase singularity behavior is visible when channel is present.

Fig. R8 Biosensing data for sample 1. Ellipsometry parameters (a) ψ and (b) Δ versus wavelength before (black curve) and after (red curve) exposing 1pM streptavidin (at 70°). Ellipsometry parameters (c) ψ and (d) Δ versus incident angle before (black curve) and after (red curve) exposing 1pM streptavidin (at 710 nm).

The measured spectra (black curves) of 0.1 mM biotin-thiol covered samples are shown in Fig. R8, Fig. R9 and Fig. R10. As can be seen, the lowest ψ_{min} (a point of non-complete darkness) is obtained at higher incident angle (above 70°) and the wavelength corresponds to ψ_{min} is drastically blue shifted to below 730 nm. It is well known that resonance wavelengths blue shift with increasing angle of incidence (Fig. R1). This is also due to the presence of a relatively rough thin layer of biotin-thiol. Also, in the case of top Au layer sample (Fig. R7), ψ_{min} is obtained at 718 nm due to lossy rough surface of Au layer.

Fig. R9 Biosensing data for sample 2. Ellipsometry parameters (a) ψ and (b) Δ versus wavelength before (black curve) and after (red curve) exposing 1pM streptavidin (at 71°). Ellipsometry parameters (c) ψ and (d) Δ versus incident angle before (black curve) and after (red curve) exposing 1pM streptavidin (at 710 nm).

As shown in Fig. R8 to Fig. R10, the spectral wavelength and angle at which lowest ψ_{min} obtained is varied from sample 1 to sample 3. This is due to the slight variation in surface roughness after functionalizing the samples with biotin-thiol. The lowest ψ_{min} obtained for sample1, sample2 and sample 3 are 1.78° at 710 nm and 70° (Fig. R8), 1.74° at 710 nm and 71° (Fig. R9) and 1.03° at 730 nm and 73° (Fig. R10), respectively. It shows that ψ_{min} increased and phase singularity decreased after functionalizing the samples with biotin-thiol. Since it is not directly possible to use the point of darkness for sensing purpose due to the absence of light and

abrupt phase jump, the obtained close to zero reflection condition ($\psi_{min}=1^\circ$ to 3°) is sufficient for biosensing.

Fig. R10 Biosensing data for sample 3. Ellipsometry parameters (a) ψ and (b) Δ versus wavelength before (black curve) and after (red curve) exposing 1pM streptavidin (at 73°). Ellipsometry parameters (c) ψ and (d) Δ versus incident angle before (black curve) and after (red curve) exposing 1pM streptavidin (at 730 nm).

The ellipsometry parameters recorded after exposing 1 pM streptavidin are shown as red curves in Fig. R8 to Fig. R10. Almost same spectral sensitivity (wavelength red shift) in p -polarized reflection spectra and ψ spectra are obtained (Fig. R8b, Fig. R9b and Fig. R10b), which represents the refractive index change due to binding of streptavidin molecules on the sensor surface. For all three samples, the obtained ψ_{min} shift is within 1.7° to 2° in both wavelength and

angular scan. However, the phase shift varies from 21° to 26° for sample 1 to sample 3. It is evident that significant phase shift is obtained at ψ_{min} point in comparison to wavelength and ψ_{min} shift. If one can carefully look for the phase shift obtained for the sample 1 to sample 3, it is clear that phase shift increases with decreasing ψ_{min} value. That means, sample 3 provides maximum response because the lowest ψ_{min} (1.03°) was recorded for sample 3. We have recorded a maximum Δ change of $\sim 26^\circ$ at 73° angle of incidence. Therefore, we can also say that maximum phase shift is possible when ψ_{min} is obtained at higher angle of incidence. Since the lowest ψ_{min} is obtained at 73° for sample 3, it provides maximum phase shift whereas the corresponding angle for sample 2 and sample 1 are 71° and 70° , respectively. Since the resonance wavelengths, ψ_{min} and angle of incidence slightly varies from sample to sample, it is not possible to include error bar in the phase sensitivity data of a single sample (measurement). Therefore, we would like to present these data as individual figures.

This section has been included in the supplementary file (Section 4)

Comment 4: *The authors introduce the state of the art in biosensing in terms of device sensitivity reported in fg/mm^2 area sensitivity. In their own experiments, they report the detection of streptavidin from a solution with pM concentration. From this concentration measurement it is not possible to infer their device sensitivity and compare with the state of the art. Related to this point, picomolar detection sensitivity for streptavidin depends on a number of parameters, including the number density of biotin receptors at the sensor surface, percentage unspecific binding, incubation time, refractive index changes due to exchange of buffer solutions, etc; parameters most of which are not reported in this manuscript.*

Author Reply: We have clarified and addressed this critical point by introducing a simple theoretical approach, which we believe delineates the device sensitivity of the presented sensor. In order to obtain an idea about the device sensitivity, we estimate the sensitivity of the phase shift to the number of molecules adsorbed on the sensor surface. In general, for each concentration c of the biomolecule in a device, there will be a corresponding saturating shift (wavelength, angle, intensity and phase), which is due to the presence of an average equilibrium population $N(c)$ of adsorbed particles on the sensor surface, a number which we cannot directly measure. In our case, device sensitivity can be defined as $\delta\Delta(c)/N(c)$, or mean phase shift per adsorbed particle.

Since $N(c)$ is not straightforward to determine, we can estimate a reliable upper bound on this number, which is $N_{max}(c)$, the maximum number of molecules on average that can be adsorbed on the sensor surface. Since $N(c) \leq N_{max}(c)$, $\delta\Delta(c)/N_{max}(c)$ will be a lower bound on the true sensitivity $\delta\Delta(c)/N(c)$. In order to estimate $N_{max}(c)$, we consider several factors that contribute to $N(c)$ such as illuminated beam area on the sensor, adsorption on boundaries of channel volume and irreversibility of adsorption.

Since the illuminated beam diameter is around 1mm^2 , the effective sensor area is 1mm^2 . Note that only a small fraction of the total population of streptavidin molecules will end up adsorbed on the illuminated sensor area. Adsorption can occur along the sensor surface and also on the PDMS top and side surfaces of the channel. For N_{max} , we will assume that only the sensor surface is adsorbing, since any competition from the PDMS surfaces will always lead to fewer molecules on the sensor. We will also assume that the adsorbed molecules are equally distributed across the entire sensor surface, which has dimensions of $7\text{mm} \times 7\text{mm}$, an area of 49mm^2 . Hence, given a certain maximum possible adsorbed population on the surface, only a fraction 1

$\text{mm}^2 / 49 \text{ mm}^2 = 0.02$ (2%) will be in the sensing region and relevant to the phase shift. In principle the equilibrium adsorbed population N reflects the net flux of molecules binding to the sensor areas minus the flux of molecules unbinding. To obtain, N_{max} we will assume that binding is irreversible, since any unbinding events will always be lower than adsorbed number.

Now we can calculate N_{max} . Since the PDMS channel has a height of 2 mm, initially there are c ($7 \text{ mm} \times 7 \text{ mm} \times 2 \text{ mm}/1\text{L}$) $\text{M}^{-1} = 5.9 c \times 10^{19} \text{ M}^{-1}$ biomolecules in solution inside the device. In the long time limit, if all of these were to be adsorbed irreversibly on the sensor surface, on average 2% of the total would be in the sensor areas. Thus:

$$N_{max}(c) = 11.8 c \times 10^{17} \text{ M}^{-1}$$

For 1 pM streptavidin solution, $N_{max} = 11.8 \times 10^5 \text{ M}^{-1}$

Since the obtained phase shift for 1 pM streptavidin solution using 1 mM biotin attached sample is 33° (see Fig. R12), the estimated device sensitivity is, $\delta\Delta / N_{max} = 33^\circ / 11.8 \times 10^5 = 2.8 \times 10^{-5}$ degrees. The corresponding device sensitivity based on spectral wavelength shift ($\delta\lambda / N_{max}$) is $1.7 \times 10^{-6} \text{ nm}$, where $\delta\lambda = 2 \text{ nm}$. It shows that the phase sensitivity of the device is one order of magnitude higher than the spectral sensitivity, even though the direct comparison is not valid because of the unit difference.

Now, we compare this sensitivity with the state of the art (*Nat. Mater.* **12**, 304-309 (2013)). In their work, authors used plasmonic gold nanoarrays with a dimension $0.2 \text{ mm} \times 0.2 \text{ mm}$ and measurement spot area of $30 \mu\text{m}^2$. Since the channel information is not available from the paper and their sensing mechanism is based on bulk prism coupling excitation technique, we use our channel dimension to get their device sensitivity based on our approach. Since their illuminated spot area is $30 \mu\text{m}^2$, only a fraction $9 \times 10^{-4} \text{ mm}^2 / 49 \text{ mm}^2 = 1.8 \times 10^{-5}$ will be in the

sensing region and relevant to the phase shift. In the long time limit, on average 1.8×10^{-5} of the total would be in the sensor areas.

Thus:
$$N_{max}(c) = 10.6 c \times 10^{14} \text{ M}^{-1}$$

Since they have used 10 pM streptavidin solution, $N_{max} = 10.6 \times 10^3 \text{ M}^{-1}$. They have reported a phase shift of 25° for 10 pM streptavidin solution using 1 mM biotin attached sample. The estimated device sensitivity of plasmonic gold nanoarray is, $\delta\Delta / N_{max} = 25^\circ / 10.6 \times 10^3 = 2.3 \times 10^{-3}$ degree. It shows that the device sensitivity of plasmonic gold nanoarray device is two orders of magnitude higher than our case. The decrease in phase sensitivity of the proposed device is due to the broad cavity resonance and the slight increase of ψ_{min} value after immobilization of ligands. However, one of the important advantages of the proposed sensor is that it is a *lithography-free* geometry so that complex nanofabrication and detection techniques are not required. At the same time, plasmonic gold nanoarray device involves much higher level of complexity. The higher sensitivity of plasmonic gold nanoarray device is due to its ability to excite super narrow mode plasmonic resonance. Since our device is based on the principle of asymmetric Fabry-Perot cavity, the excitation of super narrow resonance mode is not straightforward. Note that both narrow resonance with high quality factor and extreme singular phase are equally important for ultra-sensitive biosensing applications.

Nevertheless, it is possible to achieve narrow resonance and singular phase by replacing Ge layer with another high index dielectric such as $\text{Ge}_2\text{Sb}_2\text{Te}_5$ (GST). Experimentally obtained ellipsometry parameters of the structure using 10 nm GST layer is shown in Fig. R11a. The thickness of other layers used in the structure are: top Ag (20 nm), MMA (522 nm) and bottom Ag (50 nm). As can be seen, here both modes show narrow resonance and singular phase behavior. It shows that singular phase depends on ψ_{min} value as well as the narrow resonance. In

contrast to Ge layer sample (Fig. R11b), extreme singularity is not achieved for both modes because the obtained $\psi_{min} > 1$. However, we still believe that it is possible to achieve narrow resonance and extreme phase singularity simultaneously by optimizing the thickness of MMA layer, GST layer and Ag layer. In addition, a tunable point of darkness condition and singular phase is possible for this geometry, by switching the phase of the GST from amorphous to crystalline. We are currently working in this direction and the results will be reported elsewhere.

Fig. R11. Experimentally determined ellipsometry parameters (ψ and Δ). (a) using 10 nm GST layer and (b) using 10 nm Ge layer.

The following text has been included in the manuscript (Page 12),

“Since the illuminated beam diameter on the sample is around 1mm^2 , the estimated streptavidin molecules (N_{max}) at 1 pM concentration is around $11.8 \times 10^5 \text{ M}^{-1}$. Therefore, the calculated device sensitivity ($\delta\Delta / N_{max}$) is 2.8×10^{-5} degrees, which is one order of magnitude higher than the spectral sensitivity, $\delta\lambda / N_{max}$ (see Supplementary Section 5). In contrast to plasmonic gold nanoarray-biosensor based on singular phase concept¹¹, the device sensitivity of the proposed sensor could be further improved. Since the higher phase sensitivity of plasmonic gold nanoarray is due to super narrow mode plasmonic resonance¹¹, it could be possible to improve the device sensitivity of proposed sensor further by narrowing the linewidth of the cavity mode (by

achieving high-quality factor mode). In fact, we have achieved narrow linewidth mode by replacing Ge layer with phase change material such as $\text{Ge}_2\text{Sb}_2\text{Te}_5$ (see Supplementary Fig. 18).”

The calculation details for the estimation of device sensitivity (Section S5) and Fig. R11 are included in the supplementary file.

As another point raised by the reviewer “Related to this point, picomolar detection sensitivity for streptavidin depends on a number of parameters, including the number density of biotin receptors at the sensor surface, percentage unspecific binding, incubation time, refractive index changes due to exchange of buffer solutions, etc; parameters most of which are not reported in this manuscript”

Indeed, the number density of biotin receptors have an important role to decide the device sensitivity, especially at lower concentrations. Since the ψ_{min} value heavily depends on the effective thickness of the functionalized molecules, the phase sensitivity of the sensor varies with number density of biotin receptors at the sensor surface. In order to study this, we have performed a detailed experimental analysis by functionalizing the sensor surface with different concentrations (0.1 mM, 1 mM and 10 mM) of biotin-thiol. In all our experiments, incubation time was kept constant. Also note that the obtained phase shift is not related to percentage of unspecific binding and refractive index changes due to exchange of buffer solutions. Kindly refer reply to comment 6 for more details.

Comment 5: *The authors state: “which is much larger than the sensitivity of plasmonic sensors that works based on spectroscopic techniques: “which spectroscopic techniques out of the many possible are the authors specifically referring to? To my knowledge plasmon resonance-based*

sensors have detection sensitivity that are now reported at the level of single atomic ions, Nature Photonics 10, 733–739.

Author Reply: We would like to apologize for our incorrect statement. We had not estimated the actual device sensitivity of the sensor. Now we have removed the respective statement from the manuscript. We totally agree with reviewer that plasmon resonance based spectroscopic sensing techniques can now detect even single atomic ions. However, it is well known that plasmonic nanosensors are statistically diffusion limited. That means, as the concentration of solute decreases, the average time required to detect an event increases exponentially. We think that this diffusion limit issue can be overcome by some extent by using the proposed sensor because our device is lithography-free. Since the active area of the device can be scalable, light beam spot with any diameter can focus throughout the active area to detect the molecule. The sensitivity to the number of adsorbed particles is not the only criterion for a good sensor, because the response of the sensor will also depend on the polarizability of the molecules, and thus molecular weight. One can use a poor sensor to detect few number of high molecular weight molecules, e.g. viruses, while not being able to detect the same number for low molecular weight biomolecules like BSA and PSA or ultra-low molecular weight biomolecules like biotin. Since the darkness point heavily depends on the phase sensitivity of the device, our sensor is more suitable for detecting small molecular weight biomolecules.

Comment 6: *Endpoint measurements are always difficult to judge: it remains unclear which part of their detection signal for streptavidin results from bulk refractive index change, unspecific adsorption, temperature drifts etc. Perhaps more convincing would be a plot for the real time detection of streptavidin, plotting the signal change as the protein accumulates at the*

sensor surface over time, and including a baseline measurement before the protein is added to the sample solution?

Author Reply: We totally agree with reviewer that a study on real-time binding of biomolecules is very important to understand the performance of a sensor device. In our case, we need to plot the phase shift change at the point of darkness as the streptavidin accumulates on the sensor surface over time. As suggested by the reviewer, we have tried to do this measurement by integrating a PDMS microfluidic channel with the sensor. Since the microfluidic channel is on the top Ag surface, we could not see the singular phase behavior due to weak signal. As already shown in reply to comment 2, the top surface has an important role to achieve lowest ψ_{min} and the singular phase behavior. Since it is not possible to perform a real-time binding experiment without microfluidics channel, we are unable to show the phase shift change as the streptavidin accumulates on the sensor surface over time.

Nevertheless, here we present another approach to understand the specific binding of streptavidin molecules on biotin functionalized surface. Since the phase sensitivity of the proposed sensor depends on ψ_{min} value, we slightly change the effective thickness of adsorbed streptavidin layer. In our approach, we keep the analyte (streptavidin) concentration as 1 pM in PBS buffer and change the functionalization biotin concentration in PBS buffer from 0.1 mM to 10 mM. It is well known in streptavidin-biotin model that the effective thickness (height) of adsorbed streptavidin layer depends on various parameters such as base binding groups, cleanliness of the surface and concentration of the functionalized molecules (*British Journal of Pharmacology* 135, 1943-1950 (2002), *Molecules* 19, 12531-12546 (2014), *Chem. Commun.*, 1721–1723 (2006) and *Applied Surface Science* 258, 6056– 6063 (2012)). Therefore, we can expect a slight increase in effective thickness with increase in biotin concentration. This slight

increase of effective thickness strongly effects the ψ_{min} value and the phase sensitivity. As mentioned in reply to comment 3, we have exposed 1 pM streptavidin in PBS for 2h and followed by rinsing with PBS to remove non-specifically bound streptavidin molecules using a detachable PDMS channel (7 x 7 x 2 mm³) with inlet and outlet. Importantly, experiment was performed without removing the biotin attached sample from the ellipsometer stage to keep the same incident beam position on the sample, so that we could extract the correct shift. After 2 h of streptavidin exposure, we have removed the channel without disturbing the sample and recorded the ellipsometry spectra and *p*-polarized reflection spectra at the corresponding angle of incidence (in wavelength scan) and wavelength (in angular scan).

The biosensing results obtained using 0.1 mM, 1 mM and 10 mM biotin concentration are shown in Fig. R10, Fig. R12 and Fig. R13, respectively. Note that we have repeated the experiments for each concentration using three samples and results of the samples in which maximum phase sensitivity obtained are shown here. It is evident from the figures that ψ_{min} is increased as the concentration of biotin is increased. The recorded ψ_{min} for 0.1 mM, 1mM and 10 mM biotin concentration are 1.03°, 2.5° and 4.1°, respectively. That means, the lowest ψ_{min} is obtained for 0.1 mM biotin covered sample because 0.1 mM biotin only add an extra layer thickness <1 nm on the bare sample. On the other hand, the surface roughness is less compared to other samples. Therefore, the phase sensitivity is better for 0.1 mM biotin covered sample because of the lowest ψ_{min} value. A slight increase of ψ_{min} value ($\delta\psi=1.9^\circ$ for 0.1 mM and $\delta\psi=5.9^\circ$ for 1 mM) after exposing 1 pM streptavidin shows the specific binding of streptavidin on the biotin sites. That means the effective thickness slightly increased after exposure.

Fig. R12 Biosensing data for 1 mM biotin attached sample. Ellipsometry parameters (a) ψ and (b) Δ versus wavelength before (black curve) and after (red curve) exposing 1pM streptavidin (at 65°). Ellipsometry parameters (c) ψ and (d) Δ versus incident angle before (black curve) and after (red curve) exposing 1pM streptavidin (at 708 nm).

Fig. R13 Biosensing data for 10 mM biotin attached sample. Ellipsometry parameters (a) ψ and (b) Δ versus wavelength before (black curve) and after (red curve) exposing 1pM streptavidin (at 75°). Ellipsometry parameters (c) ψ and (d) Δ versus incident angle before (black curve) and after (red curve) exposing 1pM streptavidin (at 696 nm).

The recorded maximum phase sensitivity after exposing 1 pM streptavidin on 0.1 mM and 1 mM biotin covered sample is 26° and 33° , respectively (see Fig. R10 & Fig. R12). Note that 1 mM biotin sample provided slightly higher phase sensitivity due to the availability of large number of biotin binding sites on the sample. Since it is well known in streptavidin-biotin system that four biotin molecules bind to one streptavidin molecule, more biotin molecules are available for streptavidin to bind in 1 mM biotin sample as compared to 0.1 mM biotin sample. The observed ψ change ($\delta\psi=5.9^\circ$) also confirms that specific binding is strong in the case of 1 mM

biotin. However, in the case of 10 mM biotin covered sample weak sensitivity is obtained because the ψ_{min} value is largely enhanced to 4.1° (Fig. R13). In addition, as the concentration increases, the possibility of multiple adsorbed molecules leads to interference effects, with each additional molecule having a decreasing impact on the phase shift.

It is also clear from the *p*-polarized reflection spectra that slightly higher spectral wavelength shift (2 nm) is obtained for 1 mM biotin attached sample as compared to 0.1 mM biotin sample (1.6 nm). These results further confirm that streptavidin specific binding increases as the biotin concentration is increased from 0.1 mM to 1 mM. Since we have obtained different Psi change and Delta change with biotin concentration, and consistent red wavelength shift and positive angular shift after exposing streptavidin, the observed sensitivity of the sensor is not due to bulk refractive index change and temperature drifts. As mentioned above, we have rinsed with PBS to remove non-specifically bound streptavidin molecules using a detachable PDMS channel. This procedure definitely helps us to avoid the unspecific adsorption to a major extent. Therefore, we would like to clarify that the phase sensitivity achieved for the proposed sensor is mainly due to specific adsorption (refractive index changes due to specific binding of streptavidin on the sensor surface). Since the demonstration of real-time binding of streptavidin over time is not so straightforward in our case, we believe that the proof-of-concept approach demonstrated here is appropriate for providing a more convincing proof of the specific binding of streptavidin. However, it could be possible to obtain the singular phase behavior by integrating a very thin PMMA microfluidic channel with the sensor. We plan on exploring these issues more fully in future experiments.

This section has been included in the supplementary file (Section 4)

We have also conducted X-ray photoelectron spectroscopy (XPS) to confirm the effective functionalization of biotin-thiol utilizing Ag-S linkage. For more details, please see Supplementary Section 3.

The following text has been included in the manuscript (Page 11 to 12),

“The measured spectra (ψ and Δ) of 1 mM biotin-thiol covered sample is shown as black curve in Fig. 4a and Fig. 4b. As can be seen, the minimum reflection (a point of non-complete darkness) is obtained at higher incident angle (65°) and the wavelength corresponds to ψ_{\min} is drastically blue shifted to 708 nm. This is because the surface roughness of the sample was slightly increased due to the ultrathin layer of biotin-thiol. The thickness of this layer slightly increases as the concentration of biotin increases (see Supplementary Section 4). We then fixed the minimum reflection wavelength at 708 nm and recorded the pair of ellipsometry parameters ψ and Δ , as a function of incident angle, which are shown as black curve in Fig. 4c, and Fig. 4d, respectively.”

“The ellipsometry parameters recorded after exposing 1 pM streptavidin are shown as red curve in Fig. 4a and Fig. 4b for wavelength scan and in Fig. 4c and Fig. 4d for angular scan. The *p*-polarized reflection spectra are also shown in the inset of Fig. 4b. It is evident that significant phase shift is obtained at ψ_{\min} (wavelength and angle) as compared to corresponding ψ and wavelength change. We recorded a maximum Δ change ($\delta\Delta$) of $\sim 33^\circ$ at 65° angle of incidence for 1 pM streptavidin concentration. Hence it could be confirmed that the obtained phase sensitivity is mainly due to specific adsorption of streptavidin on the sensor surface (refractive index changes due to specific binding of streptavidin), by conducting an experiment in which the

results of different concentrations of biotin attached samples were compared (see Supplementary Section 4). It is important to note that the phase sensitivity strongly depends on ψ_{min} value.”

Comment 7: *It would also be nice have a quantitative interpretation of their sensing signal: Can the authors not determine the effective thickness of the adsorbed streptavidin layer? Related to this: why is there a blue shift of the spectrum due to the biotin layer, if understood correctly?*

Author Reply: We thank the referee for this comment. Biotin-streptavidin system is a well-studied model system for bio-recognition events because of its stable and selective noncovalent biological binding. In this system, four biotin molecules bind to one streptavidin molecule. The effective thickness determination of biotin-streptavidin system has been widely studied using different methods (*British Journal of Pharmacology* 135, 1943-1950 (2002), *Molecules* 19, 12531-12546 (2014), *Chem. Commun.*,1721–1723 (2006) and *Applied Surface Science* 258, 6056– 6063 (2012)). It is well known that the thickness (height) of this system depends on various parameters such as base binding groups, cleanliness of the surface and concentration of the functionalized molecules. We are aware that the effective thickness of adsorbed streptavidin is 2-3 nm. Since the effective thickness determination of adsorbed streptavidin is well-studied in the literature, we did not determine the thickness of adsorbed streptavidin. Since we have used smooth bare sample, base binding groups such as biotin disulphide, concentration of functionalized biotin molecules as 0.1 mM and 1mM, and the streptavidin concentration as 1pM, the effective thickness of adsorbed streptavidin should be within 2-3 nm.

As another point raised by the reviewer, related to this: why is there a blue shift of the spectrum due to the biotin layer, if understood correctly? As mentioned above, ψ_{min} is obtained at higher incident angles and the corresponding resonance wavelengths were blue shifted, after

functionalizing the bare samples with different concentrations of biotin. This is because the surface roughness of the sample was slightly increased due to the extra thin layer of biotin-thiol. The thickness of this layer slightly increases as the concentration increases. As can be seen (Fig. R10, R12 and R13), the lowest ψ_{min} (a point of non-complete darkness) is obtained at higher incident angle (above 61°) and the wavelength that corresponds to ψ_{min} is drastically blue shifted to below 730 nm. It is well known that resonance wavelengths blue shift with increasing angle of incidence (Fig. R1). This is mainly due to the presence of a relatively rough thin layer of biotin-thiol. Also, in the case of top Au layer sample (Fig. R7), ψ_{min} is obtained at 718 nm due to rough surface of Au layer. In our experiments, we have clearly observed the surface roughness issue. In the case of bare sample (top Ag layer), we could see the incident light spot clearly. However, it was not possible to observe the light spot in the case of functionalized samples and top Au layer samples due to scattering. Also, the blue shift increased with biotin concentrations due to the slight increase of surface roughness. These observations clearly indicate that the large blue shift in the spectrum is due to the surface roughness of the samples after functionalization of biotin.

To put this aspect in proper perspective, we have amended the respective sentence in the manuscript to reflect the same,

“As can be seen, the minimum reflection (a point of non-complete darkness) is obtained at higher incident angle (65°) and the wavelength corresponds to ψ_{min} is drastically blue shifted to 708 nm. This is because the surface roughness of the sample was slightly increased due to the ultrathin layer of biotin-thiol. The thickness of this layer slightly increases as the concentration of biotin increases (see Supplementary Section 4).” (Page 11)

Comment 8: *Functionalization of the surface with biotin seems to have effect on zero-point darkness measurements, degrading performance of the sensing system. Can the authors comment?*

Author Reply: The referee is absolutely correct. The performance of the sensing system degrades after functionalizing the samples with biomolecules. This is because the surface roughness of the sample slightly increases due to the extra thin layer of biotin-thiol. The thickness of this layer slightly increases as the concentration increases. However, this slight increase of ψ_{min} does not affect the biosensing applications because close to zero reflection condition ($\psi_{min}=1^\circ$ to 3°) is sufficient for biosensing. In biosensing experiments, phase sensitivity strongly depends on ψ_{min} value. Therefore, ultra-thin layer of biomolecule functionalization is required for biosensing applications for which we chose small biomolecule such as biotin (molecular weight=244 Da) for functionalizing the sensor surface. In our biosensing experiments, different concentrations (0.1 mM, 1 mM and 10 mM) of biotin was used to functionalize the sensor surface to check the ψ_{min} value and corresponding phase sensitivity. As shown in reply to comment 6, ψ_{min} increases as the concentration of biotin is increased from 0.1 mM ($\psi_{min}= 1.03^\circ$) to 10 mM ($\psi_{min}= 4.1^\circ$). Also, note that this is a common issue for this type of samples (*Nat. Mater.* **12**, 304-309 (2013)).

Comment 9: *It seems that the figure of merit they apply for zero reflection measurement does not directly apply for sensing experiments since “Therefore, close to zero reflection condition is sufficient for sensing applications”. Can the authors clarify?*

Author Reply: Yes, the FOM obtained for bare sample cannot be achieved after functionalization of the sample with biomolecules. Since ψ_{min} increases with functionalization, the FOM decreases because $FOM= 1/\psi_{min}$. Please see reply to comment 8 for more details.

Reply to Reviewer-3

Reviewer#3

I have carefully read the manuscript from Kandammathe Valiyaveedu Sreekanth et al. entitled "Realization of singular phase and point of darkness in optically thin metal-dielectric films for ultrasensitive biosensing applications". This paper uses the concept of "point of darkness" for sensing purpose, and realize the "point of darkness" using lithography-free thin film structures. The idea of this work is interesting, but I feel the overall quality of this work does not meet the standards of Nature Communications.

Reply: We would like to thank the reviewer for the time and efforts dedicated in reviewing our manuscript and providing very valuable feedback and comments. We have clarified and addressed all the critical points raised by the reviewer. Here, we have provided detailed answers to the comments and used the feedback to revise the manuscript accordingly. We have fabricated new set of samples and performed detailed experimental analysis which address the major concerns related to the novelty of the work. Therefore, we now believe that the overall quality of the paper is improved and suitable for publication in *Nature Communications*.

Comment 1: *The authors missed quite a lot of relevant existing work in the literature. In the abstract, it is said (Page 2 line 26-28), "Here, we propose and experimentally demonstrate an entirely different approach to realizing point of darkness and phase singularity using a simple metal-dielectric multilayer thin film stack." As well as in the Discussion part (Page 11 line 235-237): "The point of darkness so far has only been demonstrated based on the concept of topological darkness for which requires complex lithography techniques for fabricating nanostructures to engineer the effective index." These arguments are either too strong or not*

right. In fact, there are a lot of work been done in the literate to realized perfect absorption just using thin film structures. For example: [1] Kats, Mikhail A., et al. "Ultra-thin perfect absorber employing a tunable phase change material." *Applied Physics Letters* 101.22 (2012): 221101. [2] Li, Zhongyang, Serkan Butun, and Koray Aydin. "Large-area, lithography-free super absorbers and color filters at visible frequencies using ultrathin metallic films." *ACS Photonics* 2.2 (2015): 183-188.

The statement from this work "has only been demonstrated..." is not correct at all. The authors need to change their statement in this work, also mention the relevant work on perfect absorption using thin film structures.

Author Reply: We sincerely apologize for not citing relevant existing works in the literature. We do agree that there are some nice experimental (*APL* 101, 221101(2012) and *ACS Photonics* 2, 183(2015)) and theoretical (*Opt. Exp.* 21, 25307 (2013) and *Opt. Exp.* 22, 8339 (2014)) works which discuss about the perfect absorption of light using lithography-free thin film multilayer stacks. However, to our best of knowledge, *the complete suppression of light ($R=0$) and the phase singularity at the point of darkness using optically thin multilayer structure is not yet realized.* Importantly, we have experimentally achieved lowest psi value, $\psi_{min}=0.13^\circ$ ($R=0.001\% \neq 0$) and a large phase change of 122° using an optically thin metal-dielectric multilayer film. Therefore, the obtained figure-of-merit (FOM) of the proposed *lithography-free* system at optical wavelength (814 nm) is 445, which is 2.2 times higher than the previously reported FOM (FOM of 200 [where $\psi_{min}=0.3^\circ$] has been realized using *nanopatterned* plasmonic gold nanoarrays, *Nat. Mater.* 12, 304-309 (2013)). We have further exploited this phase singularity property at the point of darkness for the phase-sensitive sensing of biomolecules at lower concentrations, which is not demonstrated so far using thin metal-dielectric film stacks.

As the reviewer pointed out, the two published experimental works (*APL 101, 221101(2012)* and *ACS Photonics 2, 183(2015)*) have reported the perfect absorption using thin film structures. The authors of *APL 101, 221101(2012)* reported the perfect absorption of light (99.75%) at far infrared wavelength (11.6 μm) and higher temperature (343 K) using a single lossy dielectric layer on an opaque substrate. Also, the authors of *ACS Photonics 2, 183(2015)* elegantly demonstrated a narrow bandwidth (~ 17 nm) super absorber at visible wavelength with 97% maximum absorption with a performance comparable to nanostructure/nanoparticle-based super absorbers using a lithography-free asymmetric metal–insulator–metal (MIM) based Fabry–Perot cavity. However, no phase singularity behavior is reported in both papers. In the present work, we report a wide angle perfect absorption at optical wavelength and room temperature, with a maximum of 99.999% at 814 nm wavelength (See Fig. R4). Importantly, we show an abrupt phase change at the maximum absorption (minimum reflection) point, which is useful for biosensing applications. It should be noted that our proposed geometry without Ge layer even provides wide angle perfect absorption with a maximum of 99.86% at 830 nm (See Fig. R4), but phase singularity is not at all achieved. It shows that phase singularity is only possible when the multilayer film consists of a very thin layer of highly absorbing dielectric such as Ge. In the following sections, we present a detailed explanation for this interesting phase singularity behavior of the proposed sample by comparing the results of various samples (‘with’ and ‘without’ Ge layer).

We have fabricated new set of samples by optimizing the thickness of each layer to achieve minimum ψ and maximum phase change. The optimized thickness of each layer in the proposed geometry is: top Ag layer (20 nm), second MMA layer (522 nm), Ge layer (10 nm) and bottom Ag layer (80 nm). Initially, we have optimized the thickness of cavity (MMA) layer to

achieve minimum ψ and the maximum phase change. For this purpose, we have first determined the thickness of MMA layer by spin coating the MMA on Si wafer at different r. p. m. Ellipsometrically determined thickness of MMA layer at 2000, 3000 and 4000 r. p. m. are \square 630 nm, \square 522 nm and \square 443 nm, respectively. Thickness of other layers in the multilayer are: top Ag layer (20 nm), Ge layer (10 nm) and bottom Ag layer (80 nm).

Fig. R1. Experimentally determined ellipsometry parameters (ψ and Δ) and p -polarized reflection spectra of samples ‘with’ and ‘without’ Ge layer. (a) to (c) for 630 nm MMA layer, (d) to (f) for 522 nm MMA layer and (g) to (i) for 443 nm MMA layer. The incident angle is fixed where the ψ_{min} is obtained for the longer wavelength mode.

In Fig. R1, we plot the experimental ellipsometry spectra (ψ and Δ) and p -polarized reflection spectra of the samples ‘with’ and ‘without’ Ge layer for different MMA layer thickness. All the spectra were plotted for an incident angle at which minimum ψ is obtained for

the longer wavelength mode. Importantly, a singular phase behavior at ψ_{min} point and a blue shift in resonance is obtained for all the thickness of MMA layer by introducing a thin Ge layer (10 nm) in the geometry. It shows that phase singularity is only possible when Ge layer is present in the geometry. One can see that the modes are blue shifted by decreasing the thickness of MMA layer and the lowest ψ value, and minimum p -polarized reflection are obtained for a MMA layer of thickness 522 nm (Figs. d to f). For 522 nm thick MMA layer, the obtained ψ_{min} for the longer wavelength mode (above 800 nm) is 3° ($R=0.008$ at 835 nm) for ‘without Ge’ sample and 0.26° ($R= 0.0000114$ at 814 nm) for ‘with Ge’ sample. In contrast to ‘without Ge’ sample, ψ_{min} is reduced 11 times (reflection reduced to 725 times) and a sharp change in phase is obtained by using ‘with Ge’ sample.

Fig. R2. Measured pair of ellipsometry parameters as a function of incident angle. (a) for ‘without Ge’ sample at 835 nm and (b) for ‘with Ge’ sample at 814 nm.

It should be noted that reduction in ψ_{min} and higher phase change at ψ_{min} is possible for angular scan as compared to wavelength scan. Therefore, in Fig. R2, we plot the ellipsometry parameters of the samples ‘without’ and ‘with’ Ge layer, as a function of incident angle. One can see that ψ_{min} is exactly obtained at the minimum incident angle for both samples (55° for ‘without Ge’ and 61° for ‘with Ge’). It is evident from the results that ‘with Ge’ sample only

shows an abrupt phase change at ψ_{min} point (obtained 122° phase change). The obtained ψ_{min} using angular scan is 0.13° (23 times reduction compared to ‘without Ge’ sample) and the calculated FOM ($1/\psi_{min}$, where ψ is expressed in radians) is 445, which is the highest value reported so far based on singular phase concept. It indicates that a thin layer of Ge plays an important role in the geometry to achieve point of darkness and phase singularity.

Fig. R3. Experimental angular and polarization depended absorption spectra of ‘without Ge’ and ‘with Ge’ samples. Absorption spectra of ‘without Ge’ sample (a) p-polarization, and (b) s-polarization. . Absorption spectra of ‘with Ge’ sample (c) p-polarization, and (d) s-polarization.

We then experimentally investigated the influence of incident angle and polarization states on absorption ($Abs=1-R$, where $T=0$) spectra of ‘without Ge’ and ‘with Ge’ samples. The

absorption spectra as a function of incident angle (20° to 80°) and polarization states (*p*- and *s*-polarization) are shown in Fig. R3. In the case of ‘without Ge’ sample, for *p*-polarization (Fig. R3a), almost perfect absorption is obtained for all the angles of incidence and narrow band perfect absorption is obtained for *s*-polarization (Fig. R3b) at lower incident angles (below 35°). For ‘with Ge’ sample (Fig. R3c), a wide band perfect absorption for longer wavelength mode at higher incident angle is obtained for *p*-polarization. However, no perfect absorption is observed for *s*-polarization (Fig. R3d) even at lower incident angles. In particular, *p*-polarized absorption increased (Fig. R4a) and *s*-polarized absorption decreased (Fig. R4b) for ‘with Ge’ sample as compared to ‘without Ge’ sample. The *p*- and *s*-polarized absorption spectra recorded at ψ_{min} angle of ‘with Ge’ and ‘without Ge’ samples are shown in Fig. R4. Since ψ_{min} is obtained at 814 nm for ‘with Ge’ sample, the measured *p*- and *s*-polarized absorption at 814 nm wavelength are 99.999% and 63%, respectively. On the other hand, *p*-polarized reflection decreased and *s*-polarized reflection increased after introducing a thin Ge layer in the sample.

Fig. R4. Absorption spectra of ‘with Ge’ and ‘without Ge’ samples at ψ_{min} angle. (a) for *p*-polarization and (b) for *s*-polarization.

This is the reason for ‘with Ge’ samples to show reduced ψ_{min} and sharp singularity as compared

to ‘without Ge’ samples because $\tan \Psi = \left| \frac{r_p}{r_s} \right|$. We have further investigated the Ge layer

thickness on achieving phase singularity at ψ_{min} . In Fig. R5, we present the wavelength and

angular scan results of ellipsometry parameters by slightly varying the thickness (8 nm-12 nm) of

Ge layer. As can be seen, lowest ψ_{min} and maximum phase change is obtained for 10 nm Ge

thickness. We noticed that phase change is only significant when the Ge thickness is within 8-12

nm. We have also investigated the phase singularity variation by changing the thickness of top

and bottom Ag film thickness. It should be noted that the top Ag film thickness should be within

18- 21 nm range to obtain significant phase singularity. However, phase singularity is observed

for different thickness of bottom Ag film (20 nm to 100 nm) and the maximum phase change is

obtained for 80 nm thick bottom Ag film (Fig. R6).

Fig. R5. Measured ψ and Δ spectra for different thickness of Ge layer. For wavelength scan (a) 8 nm, (b) 10 nm, (c) 12 nm and for angular scan (d) 8 nm, (e) 10 nm, (f) 12 nm . The corresponding incident angle and excitation wavelength is shown in each figure.

It shows that the thickness of Ag layer, MMA layer and Ge layer are crucial for realization lowest ψ_{min} and extreme phase singularity at the point of darkness. Based on our experimental analysis, the optimized thickness required to achieve lowest ψ_{min} and extreme phase singularity are: top Ag (20 nm), MMA (522 nm), Ge (10 nm) and bottom Ag (80 nm).

Fig. R6. Measured ψ and Δ spectra for different thickness of bottom Ag layer. For wavelength scan (a) 30 nm, (b) 50 nm, (c) 80 nm and for angular scan (d) 30 nm, (e) 50 nm, (f) 80 nm . The corresponding incident angle and excitation wavelength is shown in each figure.

This section has been included in the supplementary file (Section 2).

In short, the novelty of our work is highlighted in the following points:

- This is the first work which experimentally demonstrates the lowest ψ_{min} (0.13°) and record FOM (445) using *lithography-free* geometry. Previously reported FOM was 200, which was realized using *nanostructured* plasmonic gold nanoarrays [*Nat. Mater.* **12**, 304-309 (2013)].
- This is the first work which demonstrates *extreme phase singularity* at the point of darkness using *lithography-free* geometry.

- Introduces novel chemistry for thiol functionalization of biotin molecules and its immobilization on Ag-surface.
- First experimental demonstration of phase-sensitive biosensing at lowest concentration using *lithography-free* structures.

To put the present work in the proper perspective, we have amended the respective sentences in the manuscript as,

“Here, we propose and experimentally demonstrate a new approach to realize perfect absorption (99.99%) and extreme phase singularity using a simple metal-dielectric multilayer thin film stack.” (Abstract, Page 2)

“The abrupt phase change at the point of darkness so far has only been demonstrated based on the concept of topological darkness which requires complex lithography techniques for fabricating nanostructures to engineer the effective index. In this work, we have experimentally realized a new route to achieve abrupt phase change at the point of darkness in a lithography-free four layered metal-dielectric system at the Brewster angle.’ (Discussion, Page 12)

Title of the paper has been amended as,

“Realization of singular phase in optically thin metal-dielectric films for ultrasensitive biosensing applications”

As suggested by the reviewer, we have included the following text and references related to perfect absorption based on thin film structures in the manuscript:

“Even though the perfect absorption has been demonstrated at different spectral regions using various metal-dielectric thin film stacks ³⁵⁻⁴², the extreme singular phase behavior is not yet realized.” (Page 5)

35. Kats, M. A. *et al.* Ultra-thin perfect absorber employing a tunable phase change material. *Appl. Phys. Lett.* **101**, 221101 (2012).
36. Li, Z., Butun, S. & Aydin, K. Large-area, lithography-free super absorbers and color filters at visible frequencies using ultrathin metallic films. *ACS Photonics* **2**, 183-188 (2015).
37. You, J.-B., Lee, W.-J., Won, D & Yu, K. Multiband perfect absorbers using metal-dielectric films with optically dense medium for angle and polarization insensitive operation. *Opt. Exp.* **22**, 8339-8348 (2014).
38. Shu, S., Li, Z. & Li, Y. Y. Triple-layer Fabry-Perot absorber with near-perfect absorption in visible and near-infrared regime. *Opt. Exp.* **21**, 25307-25315 (2013).
39. ElKabbash, M. *et al.* Iridescence-free and narrowband perfect light absorption in critically coupled metal high-index dielectric cavities. *Opt. Lett.* **42**, 3598-3601 (2017).
40. Kats, M. A. & Capasso, F. Optical absorbers based on strong interference in ultra-thin films. *Laser & Photonics Reviews* **10**, 699 (2016).
41. Park, J. *et al.* Omnidirectional near-unity absorption in an ultrathin planar semiconductor layer on a metal substrate. *ACS Photonics* **1**, 812-821 (2014)
42. Kocer, H.; Butun, S., Li, Z. & Aydin, K. Reduced near infrared absorption using ultra-thin lossy metals in Fabry- Perot cavities. *Sci. Rep.* **5**, 8157 (2015) (Page 19)

Comment 2: *There is a logic flaw in the analysis of the reason for the perfection absorption. In Figure 1, the authors attribute the minimum of reflection to the high absorption of the*

germanium layer. I agree with the authors that in their specific design, germanium indeed contributes significantly to the perfect absorption of their thin film structures. But this statement is not general at all. The insertion of the germanium layer is not necessary in order to realize perfect absorption. In fact, one can achieve perfect absorption using the similar structure shown in Fig. 1(b), but using appropriate combination of silver and dielectric thickness (See Ref [2] above). The authors should emphasize this point in their paper.

Author Reply: We thank the reviewer for the insightful comment. We totally agree that perfect absorption can be achieved without using Ge layer. As shown in reply to comment 1, perfect absorption is possible for ‘without Ge’ sample (Fig. R3), but phase singularity is not at all possible for this sample (Fig. R1 and Fig. R2). *Please see reply to comment 1 for more details.* Since the phase singularity is required for sensing applications, this sample cannot be used for biosensing. In Fig. R1, we have showed the ellipsometry parameters by varying the dielectric (MMA) thickness, where bottom Ag layer thickness was set to 80 nm. We have further studied the ellipsometry parameters by changing the thickness of bottom Ag layer (Fig. R7). We noticed that there is no considerable singular phase behavior by changing the thickness of bottom Ag layer thickness from 20 to 100 nm. Here, MMA layer and top Ag layer thickness were set to be 552 nm and 20 nm, respectively. We confirmed that no phase singularity is possible for ‘without Ge’ sample by optimizing the combination of silver and dielectric (MMA) thickness. Therefore, the physical mechanism of achieving singular phase in the proposed system is solely due to the presence of Ge thin film.

Fig. R7. Measured ψ and Δ spectra of ‘without Ge’ sample at lowest ψ angle for different thickness of bottom Ag layer. (a) for 30 nm and (b) for 80 nm.

As suggested by the reviewer, we have amended the sentences in the manuscript as,

“For this three-layered system, ψ_{min} value can reach as low as 3° for a specific wavelength (835 nm) and angle of incidence (55°), and both modes show perfect absorption for a wide range of angles of incidence. However, singular phase behavior is non-existent for any thickness combination of MMA and bottom Ag layer (see Supplementary Section 2).” (Page 7)

Comment 3: *No one expects perfect agreement between theory and experiment, due to fabrication issues. But the agreement between theory and experimental result of this work is not “very well” (page 9 line 117) at all. For example, in Fig. 2 (a) and (b), the difference in the phase difference Δ is 40deg (experiment) VS 150deg (theory). Also the change of phase difference profile before 875nm is quite difference. Same thing in Fig. 3 (a) and (b), the agreement between theory and experiment is not good enough. In fact, considering the fact that this work is just thin film stacks, no complicated lithography is involved, one expects a much better agreement between simulation and measurement than what is shown here.*

Author Reply: We agree that the agreement between theory and experiment for Delta spectra in both scans are not very good. In our calculations, we have tried to fit the experimental Psi spectra

first (especially ψ_{min} value and corresponding resonance wavelength and angle) and as obtained Delta spectra were plotted. Please note that it is not directly possible to get the exact Psi and Delta spectra using the same thickness of each layers. Fitting is possible only if we use different layer thickness combination for both cases, which we think to be physically incorrect. Even in our new results (Fig. R8), we have obtained the same behavior. One of the reasons for this mismatch is that the quality factor of the mode in Psi spectrum are different in experiment and calculations. Note that we have obtained high quality factor mode (Fig. R8b & Fig. R8d) in our calculations, so that large phase change at ψ_{min} point is possible compared to experimental results. Since the singular phase behavior obtained at ψ_{min} point is almost same in both scans for experiment and calculations, we think that the simulated Psi and Delta spectra are acceptable. In the calculations, ellipsometrically determined optical constants (Fig. S1) of Ag, Ge, MMA and Si were used and the thickness of thin films were slightly varied from the measured values. Thickness of layers used in the calculations are:

For Fig. 2b: top Ag (21 nm), MMA (538 nm), Ge (11 nm) and bottom Ag (80 nm)

For Fig. 2d: top Ag (18 nm), MMA (532 nm), Ge (9 nm) and bottom Ag (80 nm)

Fig. R8. (a) Measured and (b) calculated pair of ellipsometry parameters as a function of wavelength at 61° angle of incidence. (c) Measured and (d) calculated pair of ellipsometry parameters as a function of incident angle at 814 nm wavelength.

The respective sentence in the manuscript has been amended and fitted layer thickness information has been included in the manuscript to clarify this important point:

“The calculated results are comparable with the experimental data. In the calculations, experimentally determined optical constants (see Supplementary Fig. 8) of Ag, Ge, MMA and Si were used and the thickness of thin films were slightly varied from the measured values to get a comparable fit. The layer thicknesses used in Fig. 2b were 21 nm, 538 nm, 11 nm and 80 nm for top Ag, MMA, Ge and bottom Ag, respectively and the corresponding thicknesses used in Fig. 2d were 18 nm, 532 nm, 9 nm and 80 nm.” (Page 9)

Comment 4: *Page 5, line 95, it is stated that*

“Brewster angle condition does not exist if the medium is an absorbing medium with complex refractive index.”

While it is also stated that later (Page 9, line 181):

“To verify that the point of darkness is obtained at the Brewster angle, ...”

These two statements are not consistent. This needs to be addressed.

Author Reply: We apologize for not being clear on this point. It is well-known that Brewster's angle is an angle of incidence at which light with a particular polarization is perfectly transmitted through a transparent dielectric surface, with no reflection. As we have clearly mentioned in the manuscript, this concept is widely used in Brewster angle microscopy that the extinction of the reflection when the p -polarized light is incident on a semi-infinite transparent medium. However, the situation is different if the medium is an absorbing medium (metal). In this case, reflectivity of p -polarized light approaches a non-zero minimum reflection at an incident angle, which is called as pseudo Brewster angle. Since our designed optical system consists of absorbing media such as silver and germanium, it is true that the Brewster angle condition does not exist anymore. Even though absorbing media are present in our system, close to zero reflection ($R=0.001\% \approx 0$) at an incident angle and frequency was obtained which is why we called this angle as Brewster angle. Actually, it is the generalization of the Brewster angle in the complex plane and multi-layered systems.

To avoid confusion, we have corrected the respective sentence in the main text as,

“To verify that the point of darkness is obtained at the Brewster angle (pseudo), we recorded the reflectance spectrum for p - and s -polarization separately.” (Page 9)

Comment 5: Page 6, line 126-127:

“The experimental ellipsometry spectra (ψ and Δ) of all the fabricated samples at the minimum reflection angle are shown in Fig.1. ”

The ellipsometry spectra are wavelength dependent. What is minimum mean? Across all wavelength? This needs to be clarified.

Author Reply: Actually, we meant that the spectra shown in Fig. 1a to Fig. 1c were recorded for an incident angle at which minimum Psi (ψ_{min}) was obtained for longer wavelength mode (second mode). To avoid confusion, we have corrected the respective sentence in the main text as,

“The experimental ellipsometry spectra (ψ and Δ) of the fabricated samples recorded for an incident angle at which minimum ψ (ψ_{min}) obtained for longer wavelength mode (second mode) are shown in Fig.1.” (Page 6)

Comment 6: *A quantitative discussion on the requirement on magnitude on ψ should be provided. How close to zero does ψ needs to be in order to see an abrupt phase change? As the authors stated in the manuscript, sensing cannot be done at exact location where reflection goes to zero. Rather most devices work on a close point where reflection is not exactly zero. But how small exactly it needs to be in order to observe this? I think a discussion on this would be helpful.*

Author Reply: As shown in Fig. R6, the extreme singular phase is only possible when $\psi_{min} < 1$. We have obtained $\psi_{min} < 1$ for bottom Ag thickness of 50 nm and 80 nm, which are 0.82° and 0.13° , respectively. However, an abrupt phase change is possible when $\psi_{min} < 2$. The measured phase change close to ψ_{min} (phase difference taken between three points with ψ_{min} as the central point) is 84° , 160° and 85° for Ge thickness 8nm, 10nm and 12 nm, respectively (Fig. R5). For

different bottom Ag thickness, the corresponding phase change is 80° , 86° and 160° for 30 nm, 50 nm and 80 nm, respectively (Fig. R6).

Fig. R9 Biosensing data for 0.1 mM biotin attached sample. Ellipsometry parameters (a) ψ and (b) Δ versus wavelength before (black curve) and after (red curve) exposing 1pM streptavidin (at 73°). Ellipsometry parameters (c) ψ and (d) Δ versus incident angle before (black curve) and after (red curve) exposing 1pM streptavidin (at 730 nm).

Fig. R10 Biosensing data for 1 mM biotin attached sample. Ellipsometry parameters (a) ψ and (b) Δ versus wavelength before (black curve) and after (red curve) exposing 1pM streptavidin (at 73°). Ellipsometry parameters (c) ψ and (d) Δ versus incident angle before (black curve) and after (red curve) exposing 1pM streptavidin (at 730 nm).

In the case of biosensing, phase sensitivity strongly depends on ψ_{min} value. Therefore, ultra-thin layer of biomolecule functionalization is required for biosensing applications. That is why we have selected small molecule such as biotin (molecular weight=244 Da) for functionalizing the sensor surface. In our biosensing experiments, different concentrations (0.1 mM, 1 mM and 10 mM) of biotin was used to functionalize the sensor surface to check the ψ_{min} value and corresponding phase sensitivity. The biosensing data obtained using three concentrations of biotin are shown in Fig. R9 to Fig. R11. As expected, lowest ψ_{min} value was

obtained for 0.1 mM biotin concentration (it will add only <1 nm thickness on the sensor surface) and ψ_{min} value increases with biotin concentration. The recorded ψ_{min} for 0.1 mM, 1mM and 10 mM biotin concentrations are 1.03° , 2.5° and 4.1° , respectively. For more details, please see Supplementary Section 4.

Fig. R11 Biosensing data for 10 mM biotin attached sample. Ellipsometry parameters (a) ψ and (b) Δ versus wavelength before (black curve) and after (red curve) exposing 1pM streptavidin (at 73°). Ellipsometry parameters (c) ψ and (d) Δ versus incident angle before (black curve) and after (red curve) exposing 1pM streptavidin (at 730 nm).

This section has been included in the supplementary file (Supplementary section 4).

Comment 7: page 9, line 175:

“the thickness of thin films was slightly varied from the measured values, in order to get a good fit.” These numbers regarding the thickness of each layer used in simulation should be provided.

Reply: Since the complex refractive index of thermally deposited films strongly depends on thickness of film and conditions of growing, the optical constants of actual thickness of thin films (i. e. Ag (20 nm & 80 nm), Ge (10 nm) and MMA) used in the sample were determined using spectroscopic ellipsometry. In our all simulations, these optical constants were used (see Fig. S1). Thickness of each layer used in the calculations are:

For Fig. 2b: top Ag (21 nm), MMA (538 nm), Ge (11 nm) and bottom Ag (80 nm)

For Fig. 2d: top Ag (18 nm), MMA (532 nm), Ge (9 nm) and bottom Ag (80 nm)

The numbers regarding the thickness of each layer used in the simulation has been included in the manuscript,

“The layer thicknesses used in Fig. 2b were 21 nm, 538 nm, 11 nm and 80 nm for top Ag, MMA, Ge and bottom Ag, respectively and the corresponding thicknesses used in Fig. 2d were 18 nm, 532 nm, 9 nm and 80 nm.” (Page 9)

Comment 8: page 9, line 178-179, “We further analyzed the point of darkness condition by changing the Ge layer and Ag layer thicknesses”. The dependence on the performance on the Ge layers is shown in the supporting materials, but the dependence on the Ag layer thickness is not shown. How the darkness point changes as the Ag layer thickness is varied?

Author Reply: We have further investigated the Ge layer thickness on achieving phase singularity at ψ_{min} . In Fig. R5, we present the wavelength and angular scan results of

ellipsometry parameters by slightly varying the thickness (8 nm-12 nm) of Ge layer. As can be seen, lowest ψ_{min} and maximum phase change is obtained for 10 nm Ge thickness. We noticed that phase change is only significant when the Ge thickness is within 8-12 nm. We have also investigated the phase singularity variation by changing the thickness of top and bottom Ag film thickness. It should be noted that the top Ag film thickness should be within 18- 21 nm range to obtain significant phase singularity. However, the phase singularity is observed for different thicknesses of bottom Ag film (20 nm to 100 nm) and the maximum phase change is obtained for 80 nm thick bottom Ag film (Fig. R6).

Please see Fig. R6 to understand the darkness point changes with bottom Ag layer thickness variation.

Fig. R6 has been included in the supplementary file (Section 2) and the respective text in the manuscript has been amended as, (Page 9)

“We further analyzed the point of darkness condition by changing the Ge layer and Ag layer thicknesses (see Supplementary Fig. 9 and Fig. 10). We found that the phase change at the reflection-less point can be drastically changed by slightly varying the thickness of Ge.”

Comment 9: *Thin film based sensor has the advantage of lithography free. But one big advantage of sensor based on nano structures is that the decay length of the localized plasmon mode is much smaller than propagating plasmon mode (20-30 nm vs few hundred nm). Looking from Fig 3 (d), the field decay length here is on the order of few hundred nm. I think some discussion on this regarding to sensing would be helpful.*

Author Reply: We do agree that localized plasmon mode (LSP) based sensors can sometimes show better sensitivity than propagating plasmon mode (SPP) sensor particularly for the

detection at single molecule level. As it well known, both SPP and LSP measure the sensitivity of the plasmon resonant frequency to small changes in refractive index adjacent to the metal-dielectric interface. LSP sensors geometrically confine electromagnetic energy absorbed from large optical cross-sections to significantly enhance local fields within 15-20 nm of the nanoparticle surface. Although reported sensitivities for LSP sensors are $\approx 2 \times 10^2$ nm/ RIU (*Langmuir* 24, 5233-5237 (2008), *Trends in Biotechnology* 29, 343-351 (2011)) due largely to smaller plasmon-active adsorptive surfaces of nanoparticles relative to planar films, detection levels are comparable to SPP sensors for single-molecule binding events (*Nanotechnology* 21, 255503 (2010)). However, SPP sensors benefit from a high sensitivity to refractive index changes of $\approx 2 \times 10^6$ response units per RIUs for a defined planar gold film area and surface coverage by a particular analyte (*IEEE sensors Journal* 10, 531 (2010)). Therefore, SPP-based sensors have higher refractive index sensitivity than LSP-based devices, due to their greater penetration depth of SPP field into the sensing medium.

As clearly point out by the reviewer, the field decay length of the proposed geometry at point of darkness condition is few hundred nanometers, which is comparable with that of SPP sensors. Therefore, the refractive index sensitivity of the proposed thin film geometry based on singular phase concept could be equivalent to SPP sensors. However, the plasmonic resonance is narrow as compared to F-P cavity resonance. Since our device is based on the principle of asymmetric Fabry-Perot cavity, the excitation of narrow resonance mode is not straightforward. Note that both narrow resonance with high quality factor and extreme singular phase are equally important for ultra-sensitive biosensing applications.

Nevertheless, it is possible to achieve narrow resonance and singular phase by replacing Ge layer with another high index dielectric such as $\text{Ge}_2\text{Sb}_2\text{Te}_5$ (GST). Experimentally obtained

ellipsometry parameters of the structure using 10 nm GST layer is shown in Fig. R12a. The thickness of other layers used in the structure are: top Ag (20 nm), MMA (522 nm) and bottom Ag (50 nm). As can be seen, here both modes show narrow resonance and singular phase behavior. It shows that singular phase depends on ψ_{min} value as well as the narrow resonance. In contrast to Ge layer sample (Fig. R12b), extreme singularity is not achieved for both modes because the obtained $\psi_{min} > 1$. However, we still believe that it is possible to achieve narrow resonance and extreme phase singularity simultaneously by optimizing the thickness of MMA layer, GST layer and Ag layer. In addition, a tunable point of darkness condition and singular phase is possible for this geometry, by switching the phase of the GST from amorphous to crystalline. We are currently working in this direction and the results will be reported elsewhere.

Fig. R12. Experimentally determined ellipsometry parameters (ψ and Δ). (a) using 10 nm GST layer and (b) using 10 nm Ge layer.

The following section has been added to fulfill the reviewer request, (Page 13)

“As it is clear from Fig. 3c and 3d, the field decay length into the sensing medium at the point of darkness condition is few hundreds of nanometers, which is comparable with that of propagating surface plasmon polaritons (SPP)²⁵. Therefore, it could be possible to obtain higher refractive index sensitivity comparable with SPP-sensors, using the singular phase concept in metal-

dielectric film stacks. At the same time, narrow resonance (high quality factor) mode is equally important to achieve higher refractive index sensitivity for this system. Since the darkness point heavily depends on the phase sensitivity of the device, this sensor is more suitable for detecting small molecular weight biomolecules.”

REVIEWERS' COMMENTS:

Reviewer #1 (Remarks to the Author):

After considering the comments of the authors on the referee's reports and their changes in the manuscript I believe this manuscript is worthy of publication in the Journal of Nature Communication. In my opinion this is a very interesting manuscript that may trigger a wave of activity.

However, I have comment on the revised manuscript:

The authors in revised manuscript claimed that they measured ellipsometric function and XPS spectra of Ag top layer and did not find any feature related to oxygen of silver. My experience with plasmonic films based on Ag demonstrates that Ag layer is quickly oxidized.

I agree with authors that the oxidation problem can be resolved by using capping layers (protected layer) such as platinum or gold (3-5 nm thickness), or optimized thickness of bimetallic silver/gold layers (gold as an outer layer).

Reviewer #2 (Remarks to the Author):

The authors have addressed all the reviewer comments.

Reviewer #3 (Remarks to the Author):

I have read the response letter as well as the updated manuscript and supporting information for this work. I have to say the authors have addressed all my concerns on the previous review and made significant improvement to the manuscript. The manuscript now has a much better quality and can be considered published in Nature Communications.

List of changes made in the revised manuscript based on the reviewer's comments

Reviewer#1

After considering the comments of the authors on the referee's reports and their changes in the manuscript I believe this manuscript is worthy of publication in the Journal of Nature Communication. In my opinion, this is a very interesting manuscript that may trigger a wave of activity.

Author Reply: We would like to thank the reviewer for the time and efforts dedicated in reviewing the revised manuscript, response letter, and supplementary file and providing positive feedback.

Reply to Reviewer-1

Comment: *The authors in revised manuscript claimed that they measured ellipsometric function and XPS spectra of Ag top layer and did not find any feature related to oxygen of silver. My experience with plasmonic films based on Ag demonstrates that Ag layer is quickly oxidized. I agree with authors that the oxidation problem can be resolved by using capping layers (protected layer) such as platinum or gold (3-5 nm thickness), or optimized thickness of bimetallic silver/gold layers (gold as an outer layer).*

Author Reply: We agree with reviewer that top silver layer can be oxidized quickly. We have discussed this issue in the previous response letter and supplementary file. As we have mentioned before, the ellipsometry spectra of bare samples were recorded two times, two days and 30 days after fabrication. A slight increase in ψ_{\min} value is obtained after 30 days, which is due to the oxidation of thin Ag surface. However, we did not identify any peaks related to oxygen in XPS analysis before and after biotin-thiol functionalization. It could be due to the

mergence of oxygen peaks with some other elemental peaks. Note that this slight increase of ψ_{\min} not at all affect the biosensing performance because close to zero reflection condition ($\psi_{\min}=1^\circ$ to 3°) is sufficient for biosensing. At the same time, reviewer has agreed with our proposal about solving oxidation issue by using capping layers (protected layer) such as platinum or gold (3-5 nm thickness), or optimized thickness of bimetallic silver/gold layers (gold as an outer layer).

We have amended the respective text in the Supplementary file as,

“We have also recorded the XPS spectra both times. However, we did not identify any peaks related to oxygen in XPS analysis before and after biotin-thiol functionalization. It could be due to the mergence of oxygen peaks with some other elemental peaks.” (Page 27)

Reviewer#2

The authors have addressed all the reviewer comments.

Author Reply: We would like to thank the reviewer for the time and efforts dedicated in reviewing the revised manuscript, response letter, and supplementary file and providing positive feedback.

Reviewer#3

I have read the response letter as well as the updated manuscript and supporting information for this work. I have to say the authors have addressed all my concerns on the previous review and made significant improvement to the manuscript. The manuscript now has a much better quality and can be considered published in Nature Communications.

Author Reply: We would like to thank the reviewer for the time and efforts dedicated in reviewing the revised manuscript, response letter, and supplementary file and providing positive feedback.